# Competition of light- and phonon-dressing in microwave-dressed Bose polarons

G. M. Koutentakis[1*], S. I. Mistakidis[2], F. Grusdt[3,4], H. R. Sadeghpour[5], and P. Schmelcher[6,7]

**1** Institute of Science and Technology Austria (ISTA), am Campus 1, 3400 Klosterneuburg, Austria

**2** Department of Physics, Missouri University of Science and Technology, Rolla, MO 65409, USA

**3** Department of Physics and Arnold Sommerfeld Center for Theoretical Physics (ASC), Ludwig-Maximilians-Universitat München, Theresienstr. 37, München D-80333, Germany

**4** Munich Center for Quantum Science and Technology (MCQST), Schellingstr. 4, D-80799 München, Germany

**5** ITAMP, Harvard-Smithsonian Center for Astrophysics, Cambridge, Massachusetts 02138, USA

**6** Center for Optical Quantum Technologies, Department of Physics, University of Hamburg, Luruper Chaussee 149, 22761 Hamburg Germany

**7** The Hamburg Centre for Ultrafast Imaging, University of Hamburg, Luruper Chaussee 149, 22761 Hamburg, Germany

*georgios.koutentakis@ist.ac.at

## Abstract

We theoretically investigate the stationary properties of a spin-1/2 impurity immersed in a one-dimensional confined Bose gas. In particular, we consider coherently coupled spin states with an external field, where only one spin component interacts with the bath, enabling light dressing of the impurity and spin-dependent bath-impurity interactions. Through detailed comparisons with ab-initio many-body simulations, we demonstrate that the composite system is accurately described by a simplified effective Hamiltonian. The latter builds upon previously developed effective potential approaches in the absence of light dressing. It can be used to extract the impurity energy, residue, effective mass, and anharmonicity induced by the phononic dressing. Light-dressing is shown to increase the polaron residue, undressing the impurity from phononic excitations because of strong spin coupling. For strong repulsions—previously shown to trigger dynamical Bose polaron decay (a phenomenon called temporal orthogonality catastrophe), it is explained that strong light-dressing stabilizes a repulsive polaron-dressed state. Our results establish the effective Hamiltonian framework as a powerful tool for exploring strongly interacting polaronic systems and corroborating forthcoming experimental realizations.

# 1 Introduction

Polaronic excitations represent a pervasive class of quasiparticles with significant implications in multiple physics areas [1]. In the field of material science, polarons are found in various technologically significant materials [2–18]. The formation, properties and interactions of polarons are pivotal in a multitude of phenomena, such as the electric conductivity of polymers [19, 20], organic magnetoresistance [21], the Kondo effect [22], and even high-temperature superconductivity [23–28]. Given their important role and the potential to elucidate the intricate quantum properties of such structures, it is desirable to simulate them in a controlled setting. As such, ultra-cold atoms representing the main platform for quantum simulation [29], have been used to study polaron physics [30–32]. Here, we focus on Bose polarons [33–38], generated when impurities interacting with an extensive bosonic gas become dressed by the elementary excitations of the latter. These can be regarded as direct atomic analogues of the Fröhlich polarons arising in semiconductors [7, 8]. Bose polarons have recently been the subject of a considerable amount of experimental [33–38] and theoretical [39–80] investigations, aiming to explicate their stationary properties and non-equilibrium quantum dynamics. Accordingly, several techniques have been deployed for tracking Bose polaron properties, including mean-field [39–52], renormalization group [53, 54], diagrammatic [55, 56], variational [57–61], quantum Monte Carlo [62–65] and multiconfigurational variational approaches [66–82], see also the recent reviews [30–32] on these quasiparticles.

In parallel to the development of quasiparticletheories, the introduction of the light-dressed atom picture has enabled the controlled manipulation of interactions between atoms and strong electromagnetic fields [83–85]. This approach postulates that the field interacts with the atomic levels altering their properties, by admixing different energy

levels, and generating light-dressed states. These are the combined eigenstates of the field and the atom, being described by the ensuing interaction Hamiltonian, and display diverse applications in atomic, molecular, and optical physics. A prominent example is coherent population trapping [86–89], where the system is transferred in a superposition state that is completely decoupled from light and hence called dark state. Dark states facilitate coherent population transfer protocols, such as the stimulated Raman adiabatic passage [90], while constituting a special case of the more general effect of electromagnetically induced transparency [91–94], accommodating distinct applications on its own [95–110]. Furthermore, light-dressed states provide an avenue for interaction control, e.g. through Rydberg dressing [111–114] or even by invoking the Rydberg blockade effect [115–117], while their lifetime is considerably longer from the bare Rydberg state that is admixed. An additional example is the field-linked states of microwave-shielded molecules [118–122], which permit the modification of intermolecular interactions and promote the formation of ground state molecular Bose-Einstein Condensates (BECs) [123].

In the context of ground state ultracold atoms, light-induced modification of atomic states via center-of-mass coupling, is a common technique for Hamiltonian engineering leading, for instance, to artificial gauge fields [124–129] and synthetic spin-orbit coupling [130–134]. These mechanisms significantly alter the single particle transport properties, while their effect on the respective interacting dynamics has been the subject of numerous studies [135–137]. Further, it is well-known that light-dressing of multi-component Bose and Fermi gases dictates the miscibility character of the mixture [138–144]. In fact, it was recently shown that an impurity can probe the mixing properties of a light-dressed (spin-1/2) bosonic environment and form magnetic Bose polarons [73], whose properties depend crucially on the strength of the light-bath dressing. This motivates the investigation of Bose polarons created by a light-dressed impurity in a scalar BEC environment. Here, it is important to explicate whether the electromagnetic field dressing is symbiotic to the polaronic dressing caused by the excitations of the BEC, or whether it modifies the properties of the polaron in a non-trivial way.

In this work, we address this question by studying the phononic polaron-dressing of a light-dressed spinor impurity immersed in an one-dimensional bosonic environment. The light-dressing of the impurity emanates from the strong Rabi-coupling of its (pseudo) spin-states by a microwave-field. In particular, we focus on the case where only one spin-state of the impurity interacts strongly with its bosonic host, while the other one remains uncoupled. Our investigation relies on the *ab initio* variational Multi-Layer Multi-Configuration Time-Dependent Hartree method for mixtures (ML-MCTDHX) [145–147], which has a remarkable track-record in addressing polaron settings [66–82]. First, the case of an attractively interacting impurity is analyzed, which as argued in [70, 71, 81, 82] provides a representative polaronic setup as long as the impurity is miscible with its environment, i.e. away from the interaction regime where temporal orthogonality catastrophe manifests in the dynamical evolution of the system [69]. We reveal that the system is well-described by a suitably constructed extended version of the effective potential model discussed in [69–82] which treats the host distribution as an external single-particle potential. By comparing the full many-body dynamical results to the aforementioned effective potential model we assess the emergent polaronic properties, namely the energy, residue, and effective mass but can also estimate momentum fluctuations caused by effects beyond the effective potential phonon-impurity dressing. It is shown that the polaron residue increases for strong coherent dressing of the impurity states. This behavior is associated to the superposition state of the impurity with almost perfectly spatially overlapping spin-$\uparrow$ and spin-$\downarrow$ components, leading to a reduction of the polaronic dressing. This can be interpreted as an effectively reduced bath-impurity interaction interaction strength. The behavior of the

residue is reminiscent to the one of radio-frequency diabatically driven impurities [82].

Turning to strong impurity-medium repulsion it is showcased that the effective potential is again adequate for describing the polaron state. Importantly, it is found that entering the strong light-impurity dressing regime a stable polaron occurs even deep within the orthogonality catastrophe regime. This gives access to the ensuing polaronic properties which were previously available solely in terms of pump-probe spectroscopy [71]. Comparisons with the effective potential approach enable the readout of the effective mass of the light-dressed polaron state. Our results can be probed through state-of-the-art experimental techniques either via an adiabatic ramp of the microwave-dressing of the impurity state or the cooling of the impurity being exposed to the spin-coupling field.

This work is structured as follows. Section 2 introduces the spinor impurity setup and important spin operators. In Sec. 3 we elaborate on different approximate effective approaches with increasing complexity, leading to the above-mentioned extended effective potential model in Sec. 3.3. Systematic comparisons of the effective potential model with the ML-MCTDHX results in the attractive polaron scanario are performed in Sec. 4. Extensions to the case of critical repulsions for phase-separation and for strong repulsions within the temporal orthogonality catastrophe regime are discussed in Sec. 5. We summarize our findings and discuss possible extensions in Sec. 6. Appendix A explains the details of our ML-MCTDHX approach and, finally, Appendix B elucidates further technical aspects of the extended effective potential model.

## 2  Spinor-impurity Hamiltonian and spin operators

We consider the stationary properties of a strongly particle imbalanced one-dimensional multicomponent atomic system with $N_B = 100$ bosons in the majority (bath) species and $N_I = 1$ spin-1/2 bosonic impurity. The impact of multiple impurities is also briefly touched upon at specific cases stated explicitly in our description below. The impurity is exposed to an external radiofrequency field that couples its spin states[1]. A weakly repulsively interacting bath is considered such that an almost perfect BEC is formed. In the following, we focus on the equal mass setting $m_B = m_I$ corresponding to different hyperfine states of the same isotope, e.g. of $^{87}$Rb, emulating the impurity spin states and the Bose gas. The mixture is confined within the same parabolic potential $\omega_B = \omega_I$ and its many-body Hamiltonian reads

$$\hat{H} = \hat{H}_{B0} + \sum_{\alpha \in \{\uparrow,\downarrow\}} \hat{H}_\alpha + \hat{H}_{BB} + \hat{H}_{BI} + \hat{H}_S, \tag{1}$$

where $\hat{H}_{B0} = \int \mathrm{d}x \ \hat{\Psi}_B^\dagger(x) \left( -\frac{\hbar^2}{2m_B}\frac{\mathrm{d}^2}{\mathrm{d}x^2} + \frac{1}{2}m_B\omega_B^2 x^2 \right) \hat{\Psi}_B(x)$ contains the kinetic and potential energies of the bath. Accordingly, $\hat{H}_\alpha = \int \mathrm{d}x \ \hat{\Psi}_\alpha^\dagger(x) \left( -\frac{\hbar^2}{2m_I}\frac{\mathrm{d}^2}{\mathrm{d}x^2} + \frac{1}{2}m_I\omega_I^2 x^2 \right) \hat{\Psi}_\alpha(x)$ encodes the same energy contributions for the spin-$\alpha \in \{\uparrow,\downarrow\}$ component of the impurity. The short-range two-body intraspecies interactions of the BEC atoms are accounted by the term $\hat{H}_{BB} = \frac{g_{BB}}{2} \int \mathrm{d}x \ \hat{\Psi}_B^\dagger(x)\hat{\Psi}_B^\dagger(x)\hat{\Psi}_B(x)\hat{\Psi}_B(x)$. Their effective strength is chosen, herewith, to be $g_{BB} = 0.5\sqrt{\hbar^3\omega_B/m_B}$ ensuring that the bosonic host is in the Thomas-Fermi regime with radius $R_{\mathrm{TF}} = 4.22 \ \sqrt{\frac{\hbar}{m_B\omega_B}}$, while the depletion of the BEC[2] is kept below

---

[1]Technically, these can be pseudo-spin transitions among different $F$ hyperfine levels. Herewith to simplify our notation and be agnostic to implementation details we refer to them simply as spin-states.

[2]According to Penrose and Onsager [148], a Bose gas is depleted if multiple eigenstates of its one-body density matrix $\rho_B^{(1)}(x,x') = \langle\Psi|\hat{\Psi}_B^\dagger(x)\hat{\Psi}_B(x)|\Psi\rangle$ are occupied. The degree of depletion refers to the sum of the occupations of all eigenstates except the dominantly populated one.

0.9%. Similarly, $\hat{H}_{BI} = g_{BI} \int \mathrm{d}x\ \hat{\Psi}_B^\dagger(x)\hat{\Psi}_\uparrow^\dagger(x)\hat{\Psi}_\uparrow(x)\hat{\Psi}_B(x)$ is the bath-impurity interaction, characterized by an effective coupling $g_{BI}$. Importantly, only the spin-$\uparrow$ impurity interacts with the bath i.e. $g_{B\downarrow} = 0$. Further, when discussing more than one impurities they are assumed to be non-interacting, namely $g_{\uparrow\uparrow} = g_{\downarrow\downarrow} = g_{\uparrow\downarrow} = 0$. The effective couplings $g_{BB}$ and $g_{BI}$ are related to the corresponding three-dimensional $s$-wave scattering lengths and the transverse confinement length [149], being thus experimentally tunable via either Fano-Feshbach [150,151] or confinement induced resonances [149].

The Rabi coupling between the impurities is introduced via $\hat{H}_S = \frac{\hbar\Delta}{2}\hat{S}_z + \frac{\hbar}{2}(\Omega_{\mathrm{R0}}\hat{S}_+ +$ h.c.), with $\Delta$ and $\Omega_{R0}$ corresponding to the detuning and the bare Rabi frequency for $g_{BI} = 0$. The spin operators $\hat{S}_\mu$, with $\mu = x, y, z$ , acquire the form

$$\hat{S}_\mu = \frac{1}{2}\sum_{\alpha,\beta\in\{\uparrow,\downarrow\}}\int \mathrm{d}x\ \hat{\Psi}_\alpha^\dagger(x)\sigma_{\alpha\beta}^\mu\hat{\Psi}_\beta(x), \qquad (2)$$

with $\sigma_{\alpha\beta}^\mu$ denoting the Pauli matrices. Notice that $\hat{H}_S$ incorporates the so-called rotating wave approximation. This is justified since the typical energy difference between the distinct hyperfine states emulating the pseudospin impurities is typically of the order of several MHz, i.e. corresponding to the microwave regime of the electromagnetic spectrum. Additionally, $\Omega_{R0} \approx \omega_B$ and $\Delta \approx \omega_B$ of at most a few kHz [138] are required in order to couple the spin dynamics with the motional degrees of freedom of the atoms.

In order to address the ground state properties of the Rabi coupled multicomponent system we deploy the *ab initio* ML-MCTDHX approach [145]. It utilizes a variationally-optimized single-particle basis for each component upon which the many-body wavefunction is expanded, for details see Appendix A. This allows, in principle, to capture all orders of system's correlations in a computationally efficient manner. Within our setup the functionality of ML-MCTDHX is further facilitated by the almost condensed state of the bosonic environment enabling its accurate description by a small number of single-particle basis states. This corroborates the feasibility of the computations with the emergent spinor impurity dynamics, which itself requires a relatively larger basis set for its reliable representation. Throughout this work we employ harmonic oscillator units, i.e. $\hbar = m_B = \omega_B = 1$, and measure the length, time and energy in units of $\sqrt{\hbar/(m_B\omega_B)}$, $\omega_B^{-1}$ and $\hbar\omega_B$ respectively.

# 3 Effective descriptions of the Rabi-coupled system

Below, we analyze the energy spectrum of light-coupled interacting impurities by gradually introducing more complicated effective approaches. We also briefly discuss the origin of the observed polaron features within ML-MCTDHX, based on the approximations incorporated in the effective model that captures their emergence. To be concrete, we use a fixed attractive impurity-medium coupling $g_{BI} = -g_{BB} = -0.5\ \sqrt{\frac{\hbar^3\omega_B}{m_B}}$, which according to our previous works [70,71,81,82] is a representative case of the stable attractive Bose polaron in one-dimension.

## 3.1 Induced energy crossings and role of polaron interactions

It is expected that the Rabi coupling term, $\hat{H}_S$, plays a crucial role in determining the many-body ground state of the Bose gas hosting $N_I$ impurities. For $\Omega_{R0} = 0$, it holds that $[\hat{S}_z, H] = 0$ and therefore the lowest-in-energy eigenstate for each value of $S_z =$

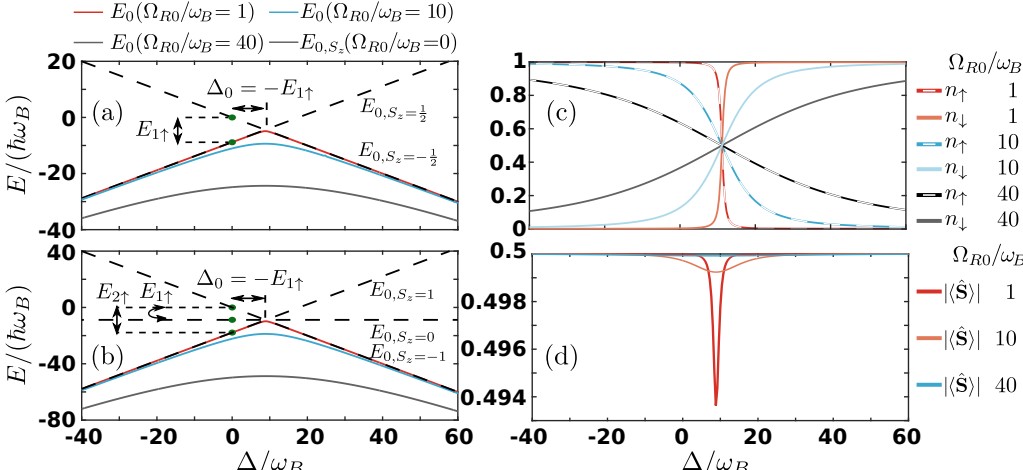

Figure 1: Description of Rabi-coupled polarons in terms of a two level model and its limitations. Ground state energies of Eq. (1) for (a) $N_I = 1$ and (b) $N_I = 2$ impurities with $g_{BI} = -g_{BB} = -0.5\ \sqrt{\hbar^3\omega_B/m_B}$ and varying $\Omega_{R0}$ (see legend). The dashed lines indicate the lowest in energy eigenstates for different $S_z$ and $\Omega_R = 0$. The important energy scales, $E_{1\uparrow}$, $E_{2\uparrow}$ and $\Delta_0$ are also schematically illustrated. (c) Population of the spin-$\uparrow$ and spin-$\downarrow$ states and (d) expectation values of the spin-magnitude $|\langle \hat{S}_\mu \rangle|$ for varying detuning $\Delta$ and for $N_I = 1$ at different Rabi-couplings $\Omega_{R0}$ (see legend). The polaronic (non-interacting) impurity states are reproduced for $\Delta \to -\infty$ ($\Delta \to \infty$). For $\Delta \approx -E_{1\uparrow}/\hbar$, a correlated superposition state is created.

$-\frac{N_I}{2}, -\frac{N_I}{2} + 1, \ldots, \frac{N_I}{2}$ turns out to be

$$\left| \Psi_{0;S_z=-\frac{N_I}{2}+n} \right\rangle = \frac{\left(\hat{a}_{0\downarrow}^\dagger\right)^{N_I-n}}{\sqrt{(N_I-n)!}} \left| \Psi_{B+n\uparrow} \right\rangle. \tag{3}$$

Here, $|\Psi_{B+n\uparrow}\rangle$ denotes the ground state of the $(N_B + n)$-body system consisting of the bath ($N_B$ atoms) coupled to $n$ interacting spin-$\uparrow$ impurities. Since $g_{B\downarrow} = g_{\uparrow\downarrow} = 0$, the spin-$\downarrow$ impurities are un-correlated with the remainder of the system. Also, the operator $\hat{a}_{0\downarrow}^\dagger$ creates a spin-$\downarrow$ boson in the ground-state (index "0") of the parabolic trap. The respective eigenenergies of the entire $(N_B + N_I)$-body system read

$$E_{0;S_z=-N_I/2+n}(\Delta) = \langle \Psi_{B+n\uparrow} | \hat{H} | \Psi_{B+n\uparrow} \rangle + \frac{\hbar\Delta}{2} S_z + (N_I - n) \frac{\hbar\omega_B}{2}. \tag{4}$$

The interaction energy of $n$ polarons refers to $E_{n\uparrow} - nE_{1\uparrow}$, where $E_{n\uparrow} \equiv \langle \Psi_{B+n\uparrow} | \hat{H} | \Psi_{B+n\uparrow} \rangle - \langle \Psi_{B+0\uparrow} | \hat{H} | \Psi_{B+0\uparrow} \rangle - n\hbar\omega_B/2 = E_{0;-N_I/2+n}(0) - E_{0;-N_I/2}(0)$ and $E_{1\uparrow}$ is the energy of a single polaron. Accordingly, for a single impurity the two energies, i.e. $E_{0;\pm1/2}$, cross for $\Delta_0 = -E_1$, see Eq. (4). Since typically Bose polarons interact, we expect multiple exact crossings (for $\Omega_{R0} = 0$), among the distinct $S_z$ states, which tend to coincide into a single one when polaron interactions become negligible, i.e. $E_{n\uparrow} \to nE_{1\uparrow}$. A concrete example with the ML-MCTDHX approach is provided by the dashed lines in Fig. 1(a) and 1(b) for $N_I = 1$, $N_I = 2$ respectively, forming an attractive Bose polaron at $g_{BI} = -g_{BB} = -0.5\sqrt{\hbar^3\omega_B/m_B}$. The presence of weak attractive induced interactions manifests by the fact that $E_{2\uparrow} \approx -17.82\hbar\omega_B$ is $\sim 1\%$ smaller than $2E_{1\uparrow} \approx 2 \times (-8.82)\hbar\omega_B$. Notice that in the case of $N_I = 2$ illustrated in Fig. 1(b), two distinct exact crossings appear that are not resolved in the figures.

In contrast, a finite $\Omega_{R0}$ leads to the coupling among the $|\Psi_{0;S_z}\rangle$ states and the emergence of an avoided crossing [85], compare the ground state energies, $E_0$, for $\Omega_{R0} \neq 0$ with $E_{0;S_z}$ for $\Omega_{R0} = 0$, represented in Fig. 1(a),(b) by solid and dashed lines respectively. Independently of $N_I$ and for $\Omega_{R0} \neq 0$, $\Delta < -E_{1\uparrow} - 2\Omega_{R0}$, it holds that $E_0 \approx E_{0;Sz=-N_I/2}$, with the latter being the overall ground state for $\Omega_{R0} = 0$ in this $\Delta$ range, while for $\Delta > -E_{1\uparrow} + 2\Omega_{R0}$ we obtain $E_0 \approx E_{0;Sz=+N_I/2}$. Substantial deviations among $E_0$ and $E_{0;S_z}$ occur for $|\Delta + E_{1\uparrow}| < 2\Omega_{R0}$ with $E_0 < E_{0;S_z}$ due to the coupling of the $S_z = \pm 1/2$ states caused by $\Omega_{R0} \neq 0$. The main difference among the energies of the $N_I = 2$ (Fig. 1(b)) and the $N_I = 1$ (Fig. 1(a)) settings is that in the former case the magnitude of the contributing energies is twice larger than the former, since also $S_z$ is increased in the same way. However, since $E_{1\uparrow} \approx 2\Omega_{R0}$ the energy scale of light-dressing is orders of magnitude larger than the polaron-polaron interactions. For this reason in the remainder of this work we will predominantly focus on the $N_I = 1$ case. Finally, notice that strong impurity-impurity interactions (when compared to $\hbar\omega_B$) might be possible for $g_{BI} \to -\infty$ [152]. However, since we cannot reliably address numerically this case within our approach we will not further discuss implications in this limit.

## 3.2 Effective two-level system

Considering a weak Rabi-coupling, i.e. $\Omega_{R0} \ll \omega_B$, it is natural to assume that the ground state, $|\Psi_0\rangle$, of the spinor system described by Eq. (1), is a linear superposition of the spin-$S_z$ eigenstates $|\Psi_{0;S_z}\rangle$ for $\Omega_{R0} = 0$. Moreover, within the bath-impurity interaction regime where polaron states are supported, the ground state of the spin-$\downarrow$ atoms, $|\Psi_{0,-1/2}\rangle = \hat{a}_{0\downarrow}^{\dagger}|\Psi_{B+0\uparrow}\rangle$, predominantly couples via $\Omega_{R0}$ with the corresponding ground state of the polaron, $|\Psi_{0;+1/2}\rangle = |\Psi_{B+1\uparrow}\rangle$, due to their large overlap which defines the polaron residue, $Z = |\langle\Psi_{B+0\uparrow}|\hat{a}_{0\uparrow}|\Psi_{B+1\uparrow}\rangle|$. In this sense, the ground state of the entire Rabi-coupled setting for $0 \neq \Omega_{R0} \ll E_{1\uparrow}$, can be approximated by a two level system involving only the non-quasiparticle $|\Psi_{B+0\uparrow}\rangle$, and polaron $|\Psi_{B+1\uparrow}\rangle$ states,

$$\hat{H}^{2\text{lvl}} = \left(E_{1\uparrow} + \frac{\hbar\Delta}{2}\right)|\Psi_{0;1/2}\rangle\langle\Psi_{0;1/2}| - \frac{\hbar\Delta}{2}|\Psi_{0;-1/2}\rangle\langle\Psi_{0;-1/2}|$$
$$+ \frac{\hbar Z}{2}\left(\Omega_{R0}^*|\Psi_{0;-1/2}\rangle\langle\Psi_{0;1/2}| + \Omega_{R0}|\Psi_{0;1/2}\rangle\langle\Psi_{0;-1/2}|\right). \tag{5}$$

The physical motivation behind this approximation is that we neglect the light-induced modification of the non-interacting and polaronic impurity states. Diagonalizing the Hamiltonian given by Eq. (5) yields the ground state of the two-level system

$$E_0^{2\text{lvl}} = \frac{1}{2}\left(E_{1\uparrow} - \sqrt{(E_{1\uparrow} + \hbar\Delta)^2 + \hbar^2|\Omega_{R0}|^2Z^2}\right), \tag{6}$$

while the populations of the $\Omega_{R0} = 0$ eigenstates, $|\Psi_{0;S_z}\rangle$, in the $\Omega_{R0} \neq 0$ ground state, $|\Psi_0\rangle$, read

$$|\langle\Psi_{0;\mp 1/2}|\Psi_0\rangle|^2 = \frac{1}{2}\left[1 \pm \frac{\text{sign}(E_{1\uparrow})(E_{1\uparrow} + \hbar\Delta)}{\sqrt{(E_{1\uparrow} + \hbar\Delta)^2 + \hbar^2|\Omega_{R0}|^2Z^2}}\right]. \tag{7}$$

Finally, the relative phase of the resulting superposition is solely dictated by $\Omega_{R0}$ namely

$$\frac{\langle\Psi_{0;-1/2}|\Psi_0\rangle}{|\langle\Psi_{0;-1/2}|\Psi_0\rangle|} = -e^{\arg(\Omega_{R0})}\frac{\langle\Psi_{0;+1/2}|\Psi_0\rangle}{|\langle\Psi_{0;+1/2}|\Psi_0\rangle|}. \tag{8}$$

To establish the validity of this simplified approach, we present in Fig. 1(a) and 1 (c) the energies and populations of the spin-states, $n_{\uparrow,\downarrow} = |\langle\Psi_{0;+1/2,-1/2}|\Psi_0\rangle|^2$, respectively

within the fully correlated approach in the case of $N_I = 1$. The ground state energies [Fig. 1(a)] are in excellent agreement with the two-level approximation predictions [Eq. (6), Eq. (7)] which are not depicted since they are almost indistinguishable from the ML-MCTDHX ones in the presented scales. We remark that by employing the ML-MCTDHX calculated polaron residue $Z = |\langle \Psi_{0;1/2}|\Psi_{0;-1/2}\rangle| = 0.984$ in Eq. (6), the maximum observed deviation between the two-level and the ML-MCTDHX approaches is $\sim 1\%$ at $\Omega_{R0} = 40\omega_B$. Similarly, the behavior of the spin state populations, $n_\alpha$, with respect to $\Delta$ is almost perfectly described within the two-level assumption. It can be clearly seen that the spin-$\uparrow$ state becomes fully occupied for larger $\Delta/\omega_B$ due to the increasing fictitious magnetic field polarizing the impurity. Notice also that for small values of the polaron interaction energy, $E_{n\uparrow} - nE_{1\downarrow}$, the system is SU(2) invariant and thus the $N_I > 1$ extension is trivial. Hence, it can be proved that the ground state of the $N_I$ system takes the form $\hat{R}(\Delta, \Omega_{R0})|\Psi_{0;-N_I/2}\rangle$, where $\hat{R}(\Delta, \Omega_{R0}) = \exp(i\hat{S}_y \cos^{-1} \frac{\text{sign}(E_{1\uparrow})(E_1 + \hbar\Delta)}{2\sqrt{(E_{1\uparrow} + \hbar\Delta)^2 + \hbar^2|\Omega_{R0}|^2 Z^2}})$ is the spin rotation operator, which is also in excellent agreement with the ML-MCTDHX results for $N_I = 2$.

The deviations among the fully correlated system and the effective two-level description tion can be elucidated by considering the spin-magnitude $|\langle \hat{\boldsymbol{S}} \rangle| = \sqrt{\sum_{\mu=x,y,z}\langle\Psi_0|\hat{S}_\mu|\Psi_0\rangle^2}$. Using the fact that within the two-level model the state of the system is a linear combination of $|\Psi_{0;\pm 1/2}\rangle$ and taking into account the spin populations [Eq. (7)] and the phase of the superposition [Eq. (8)], we obtain

$$|\langle \hat{\boldsymbol{S}} \rangle| = \frac{1}{2}\sqrt{\frac{(E_{1\uparrow} + \hbar\Delta)^2 + \hbar^2|\Omega_{R0}|^2 Z^4}{(E_{1\uparrow} + \hbar\Delta)^2 + \hbar^2|\Omega_{R0}|^2 Z^2}}. \tag{9}$$

The latter has a minimum at resonance $\Delta = -E_{1\uparrow}/\hbar$ with value $|\langle \hat{\boldsymbol{S}} \rangle|_{\min} = Z/2$, implying that, for the parameters employed in Fig. 1, $|\langle \hat{\boldsymbol{S}} \rangle|_{\min} = 0.492$ for every $\Omega_{R0}$ within the two-level approximation. However, the ML-MCTDHX data, see Fig. 1(d), follow the same functional form as in Eq. (9) but with a larger value of $Z = Z_{\text{eff}}(\Omega_{R0})$. The increase of $Z$ in the ML-MCTDHX data can be verified by observing that $|\langle \hat{\boldsymbol{S}} \rangle|_{\min} > 0.492$ increases with $\Omega_{R0}$. This modification can be interpreted in two ways, namely: i) either as a genuine reduction of the impurity dressing owing to the light-matter dressing, or ii) a modification of the impurity state close to resonance due to $\Omega_{R0} \neq 0$ resulting in a larger overlap between the spin-states. To discern between these two modification mechanisms of the polaronic dressing, in the following section, we develop an effective model by treating the BEC as a material barrier for the impurity [69–82].

## 3.3 The extended effective potential approach

It was recently demonstrated [69–82] that effective one-dimensional potential approaches neglecting impurity-bath correlations can be employed to qualitatively address the properties and stability of the Bose polaron. To derive an effective single-particle Hamiltonian, describing a Rabi coupled impurity inside a BEC, we consider that only the spin-$\uparrow$ impurity experiences an effective potential owing to its interaction with the bosonic host. Within this framework, the effects due to the phononic dressing of the impurity are phenomenologically taken into account by the following assumptions: (i) the mass of the spin-$\uparrow$ atom is renormalized to the effective mass, $m_I^*$. (ii) There is an energy correction of the single-particle model $\delta E_p \equiv E_{1\uparrow} - \langle\psi_{0\uparrow}|\hat{H}_{\text{eff}}|\psi_{0\uparrow}\rangle + \langle\psi_{0\downarrow}|\hat{H}_{\text{eff}}|\psi_{0\downarrow}\rangle$, where $E_{1\uparrow}$ is the exact polaron energy. (iii) A correction due to the modification of the bath states owing to the impurity dressing, i.e. corresponding to phononic excitations, is introduced.

The energy correction (ii) is unambiguous and thus simple to implement. However, the effective mass (i) and phononic (iii) corrections are more involved. In the homogeneous case, $\omega_B = 0$, the effective mass corresponds to the second derivative of the polaron dispersion, namely $m_I^* = (\frac{\mathrm{d}^2 E_{1\uparrow}}{\mathrm{d}p^2})^{-1}$. However, for a trapped system the polaron momentum is not a good quantum number and hence $m_I^*$ is not directly available from comparison with experiment or calculations. It has been proposed that $m_I^*$ can be estimated through analyzing the impurity collective modes e.g. its breathing or dipole dynamics in comparison to effective models [68,70]. In this spirit, here, we consider $m_I^*$ as a fitting parameter of our model to be evaluated via its comparison with the numerically obtained ML-MCTDHX data.

Since the effective potential model neglects bath-impurity correlations, it makes sense to assume a tensor product ansatz $|\psi_{n\sigma}\rangle \approx |\psi_{n\sigma}^B\rangle \otimes |\psi_{n\sigma}^I\rangle$, where $|\psi_{n\sigma}^B\rangle$ and $|\psi_{n\sigma}^I\rangle$ represent the bath and the impurity states respectively characterized by the spatial, $n$, and spin, $\sigma$, indices. Notice here that the product ansatz does not imply that the state of the bath is independent of the state of the impurity but rather that it parametrically depends on the impurity state via the index $n$. Then, since the states of the impurity can be calculated within the effective potential approach, the overlap of the bath states can be fixed such that the exact polaron residue corresponding to the ground state of the impurity, $n = 0$, is reproduced i.e. $Z_{\mathrm{eff}} = \langle \psi_{0\uparrow}^B | \psi_{0\downarrow}^B \rangle = Z / \langle \psi_{0\uparrow}^I | \psi_{0\downarrow}^I \rangle$ for $\Omega_{\mathrm{R}0} = 0$.

With these assumptions the effective Hamiltonian, $\hat{H}_{\mathrm{eff}}$, experienced by the impurity reads

$$
\begin{aligned}
\hat{H}_{\mathrm{eff}} = &\left( -\frac{\hbar^2}{2m_I^*}\hat{P}_\uparrow - \frac{\hbar^2}{2m_I}\hat{P}_\downarrow \right) \frac{\mathrm{d}^2}{\mathrm{d}x^2} + \frac{1}{2}m_I \omega_I^2 x^2 + \left( g_{BI}\rho_B^{(1)}(x) + \delta E_p \right)\hat{P}_\uparrow \\
&+ \hbar\Omega_{\mathrm{R}0}\hat{S}_x + \hbar\Delta\hat{S}_z,
\end{aligned}
\tag{10}
$$

where $\hat{P}_\sigma = \int \mathrm{d}x \, \hat{\Psi}_\sigma^\dagger(x)\hat{\Psi}_\sigma(x)$ are the projectors to the $\sigma \in \{\uparrow,\downarrow\}$ impurity spin state. Notice that the expectation values $\langle \psi_{n\alpha}|\hat{P}_\sigma|\psi_{n\beta}\rangle = \langle \psi_{n\alpha}^I|\hat{P}_\sigma|\psi_{n\beta}^I\rangle$ and $\langle \psi_{n\alpha}|\hat{S}_z|\psi_{n\beta}\rangle = \langle \psi_{n\alpha}^I|\hat{S}_z|\psi_{n\beta}^I\rangle$ are equal when acting to both the total state or the impurity state, since they correspond to diagonal operators in the spin-basis. However, the same is not true for $\hat{S}_x$. In this case, $\langle \psi_{n\uparrow}|\hat{S}_x|\psi_{n\downarrow}\rangle = \langle \psi_{n\uparrow}^B|\psi_{n\downarrow}^B\rangle\langle \psi_{n\uparrow}^I|\hat{S}_x|\psi_{n\downarrow}^I\rangle \neq \langle \psi_{n\uparrow}^I|\hat{S}_x|\psi_{n\downarrow}^I\rangle$. Therefore, in order to render the effective Hamiltonian of Eq. (10) bath-agnostic we renormalize the coefficients of the spin-operators as follows $\hat{S}_x \to Z_{\mathrm{eff}}/2 \, \hat{\sigma}_x$, $\hat{S}_y \to Z_{\mathrm{eff}}/2 \, \hat{\sigma}_y$ and $\hat{S}_z \to 1/2 \, \hat{\sigma}_z$, with $Z_{\mathrm{eff}} = \langle \psi_{0\uparrow}^B | \psi_{0\downarrow}^B \rangle$. The change of notation from $\hat{S}_\mu$ to $\hat{\sigma}_\mu$ serves as a reminder that this transformation should be inverted for calculating spin-dependent quantities such as $|\langle \hat{\boldsymbol{S}}\rangle|$, see further details in Appendix B. This transformation incorporates the additional assumption that the overlap of the bath states is not dependent on the state of the impurity $\langle \psi_{n\uparrow}^B | \psi_{n\downarrow}^B \rangle \approx \langle \psi_{0\uparrow}^B | \psi_{0\downarrow}^B \rangle$. This can be justified by the fact that we are mostly interested in the ground state of the effective potential and thus $Z_{\mathrm{eff}}$ can be assumed to be a function of $\Omega_{\mathrm{R}0}$, i.e. $Z_{\mathrm{eff}}(\Omega_{\mathrm{R}0}) = \langle \psi_{0\uparrow}^B(\Omega_{\mathrm{R}0}) | \psi_{0\downarrow}^B(\Omega_{\mathrm{R}0}) \rangle$ which needs to be self-consistently determined by comparisons with the ML-MCTDHX approach. As such, it is in principle a fitting parameter. However, for most of the discussion that follows we will consider $Z_{\mathrm{eff}} = 1$, since the $Z_{\mathrm{eff}}$ renormalization is found to affect only weakly our results.

To proceed let us further assume the Thomas-Fermi approximation for the state of the bath, as also confirmed by our many-body calculations in the considered parameter regime. Accordingly, the bath density profile reads

$$
\rho_B^{(1)}(x) = \begin{cases} \frac{m_B\omega_B^2}{2g_{BB}}\left(R_{\mathrm{TF}}^2 - x^2\right) & \text{if } |x| \le R_{\mathrm{TF}}, \\ 0 & \text{if } |x| > R_{\mathrm{TF}} \end{cases},
\tag{11}
$$

with the Thomas-Fermi radius $R_{\text{TF}} = \left( \frac{3 g_{BB} N_B}{2 m_B \omega_B^2} \right)^{1/3}$. Within this approximation the effective Hamiltonian is simplified to

$$
\begin{aligned}
\hat{H}_{\text{eff}} =& \frac{E_0}{2} + \hbar \tilde{\omega}_I \left( \hat{a}^\dagger \hat{a} + \frac{1}{2} \right) + \frac{\hbar \Omega_{\text{eff}}}{2} \hat{\sigma}_x + \frac{\hbar \Delta + E_0}{2} \hat{\sigma}_z \\
& - \frac{\hbar \Delta_h}{2} \left( \hat{a}^\dagger \hat{a} + \frac{1}{2} \right) \hat{\sigma}_z - \frac{\hbar \Lambda}{2} \left[ \left( \hat{a}^\dagger \right)^2 + \hat{a}^2 \right] \hat{\sigma}_z.
\end{aligned}
\tag{12}
$$

Apparently, the effective Hamiltonian, $\hat{H}_{\text{eff}}$, describes a spin-dependent harmonic confinement of the impurity, see $\bar{\omega}_I$, $\Delta_h$, with additional spin-orbit coupling provided by $\Lambda$ and Rabi-dressing dictated by $\Omega_{\text{eff}}$ and $\Delta$. This description holds provided that the impurity is not able to escape its bosonic host, i.e. it is confined in the $|x| \le R_{\text{TF}}$ spatial region. Let us postpone for the moment the discussion regarding the parametric range of validity of this approximation, in order to first explain the terms appearing in the above expression. An important quantity is the localization length scale of the impurity

$$
\ell = \sqrt{\frac{\hbar}{m_I \omega_I}} \left[ \frac{\frac{1}{2} \left( 1 + \frac{m_I}{m_I^*} \right)}{1 - \frac{1}{2} \frac{g_{BI}}{g_{BB}} \frac{m_B \omega_B^2}{m_I \omega_I^2}} \right]^{1/4},
\tag{13}
$$

via which the creation (annihilation) operators are defined as $\hat{a}^\dagger = \frac{1}{\sqrt{2}} \left( \frac{\hat{x}}{\ell} - \frac{\ell \hat{p}}{\hbar} \right)$ ($\hat{a} = \frac{1}{\sqrt{2}} \left( \frac{\hat{x}}{\ell} + \frac{\ell \hat{p}}{\hbar} \right)$). From Eq. (13), it can be verified that $\ell \le 2^{1/4} \sqrt{\frac{\hbar}{m_I \omega_I}}$, in the case that temporal orthogonality catastrophe is not observed, i.e. $g_{BI} < g_{BB} \frac{m_I \omega_I^2}{m_B \omega_B^2}$, and hence Eq. (12) is valid as long as $\sqrt{\frac{\hbar}{m_I \omega_I}} \le R_{\text{TF}}$. Having at hand these definitions, the parameters of the effective Hamiltonian can be expressed as

$$
\Omega_{\text{eff}} = \Omega_{\text{R0}} Z_{\text{eff}},
\tag{14a}
$$

$$
E_0 = \frac{1}{2} \frac{g_{BI}}{g_{BB}} m_B \omega_B^2 R_{\text{TF}}^2 + \delta E_p,
\tag{14a}
$$

$$
\tilde{\omega}_I = \omega_I \sqrt{\frac{1}{2} \left( 1 + \frac{m_I}{m_I^*} \right) \left( 1 - \frac{1}{2} \frac{g_{BI}}{g_{BB}} \frac{m_B \omega_B^2}{m_I \omega_I^2} \right)},
\tag{14c}
$$

$$
\Delta_h = \omega_I \sqrt{\frac{\frac{1}{2} \left( 1 + \frac{m_I}{m_I^*} \right)}{1 - \frac{1}{2} \frac{g_{BI}}{g_{BB}} \frac{m_B \omega_B^2}{m_I \omega_I^2}}} \left( \frac{\frac{m_I}{m_I^*}}{1 + \frac{m_I}{m_I^*}} \frac{g_{BI}}{g_{BB}} \frac{m_B \omega_B^2}{m_I \omega_I^2} + \frac{1 - \frac{m_I}{m_I^*}}{1 + \frac{m_I}{m_I^*}} \right),
\tag{14d}
$$

$$
\Lambda = \frac{1}{2} \omega_I \sqrt{\frac{\frac{1}{2} \left( 1 + \frac{m_I}{m_I^*} \right)}{1 - \frac{1}{2} \frac{g_{BI}}{g_{BB}} \frac{m_B \omega_B^2}{m_I \omega_I^2}}} \left( \frac{1}{1 + \frac{m_I}{m_I^*}} \frac{g_{BI}}{g_{BB}} \frac{m_B \omega_B^2}{m_I \omega_I^2} - \frac{1 - \frac{m_I}{m_I^*}}{1 + \frac{m_I}{m_I^*}} \right).
\tag{14e}
$$

The intuitive interpretation of Eq. (12) is that the effective potential caused by the bosonic environment modifies the frequency of the trap from its average value $\bar{\omega}_I$ in a spin-dependent manner. The magnitude of this change is given by $\Delta_h$, which appears in Eq. (12) as a state-dependent detuning. This effect in addition causes the state to be squeezed, i.e. the position and momentum uncertainty become spin-state dependent which is captured by the parameter $\Lambda$. Since $\Lambda$ and $\Delta_h$ are related to shifts of the trap length and frequency respectively their interrelation is controlled by the effective mass of the polaron $m_I^*$. Notice that $\Lambda = \frac{1}{2} \Delta_h$ for $m_I^* = m_I$, see also Eq. (14e). In the case of strong light dressing $|\Omega_{\text{eff}}| \gg |E_0/2 - \hbar \Delta_h/4|$, both of the aforementioned state dependent effects

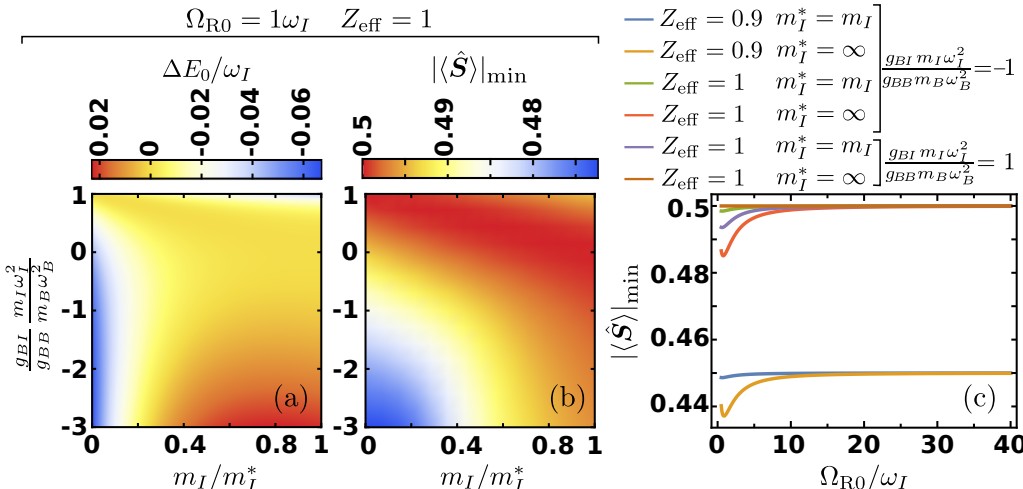

Figure 2: Deviations from the two-level system description of microwave polaron-dressing captured by the effective-potential model. (a) The energy shift owing to the spin-orbit coupling term $\propto \Lambda$ within second order perturbation theory to the effective potential of Eq. (12). (b) The minimum value of $|\langle \hat{S} \rangle|$ within the same approximation. In both cases we consider $\Omega_{R0} = 1\,\omega_I$ and $Z_{\text{eff}} = 1$. (c) The $\Omega_{R0}$ dependence of the minimum of $|\langle \hat{S} \rangle|$ for different values of $Z_{\text{eff}}$, $m_I^*$ and $\frac{g_{BI} m_I \omega_I^2}{g_{BB} m_B \omega_B^2}$ (see legend). Notice that only the region $\Omega_{R0} > 0.5\omega_I$ is presented to avoid issues associated with the breakdown of perturbation theory.

become negligible and the impurity state is well approximated by a tensor product of a spinless particle confined in a trap with strength $\bar{\omega}_I$ and a spin-1/2 atom which interacts with the field.

For $\Lambda = 0$ the Hamiltonian of Eq. (12) can be diagonalized analytically, see Appendix B.2. With the additional assumptions $Z_{\text{eff}} = Z$ and $\delta E_p = E_{1\uparrow} + \hbar\Delta_h/4 - g_{BI} m_B \omega_B^2 R_{\text{TF}}^2/(2g_{BB})$ (ensuring the resonance condition for $\Delta = -E_{1\uparrow}$) this effective potential model [Eq. (12)] is equivalent to the two-level model of Eq. (5). Even in the case of a finite value of $\Lambda$ the predictions of the effective potential approach are not drastically different from the two-level model. Moreover, it is possible to show that for $\Omega_{R0} = 0$ and away from the temporal orthogonality catastrophe regime [69], i.e. $g_{BI} < g_{BB} \frac{m_I \omega_I^2}{m_B \omega_B^2}$, the upper bound for the energy correction $\delta E_p$ is $\Delta E_{\max} = \frac{\hbar\tilde{\omega}_I}{8}(1 + m_I/m_I^*)^{-2} \leq \frac{\hbar\tilde{\omega}_I}{8}$ (see also Appendix B.1 for the detailed derivation). This enables a perturbative treatment of the spin-orbit coupling term $\propto \Lambda$ of Eq. (12) resulting to

$$\Delta E_0 = \hbar\Lambda^2 \Delta_h \frac{\Delta_h^2 - 4\hbar\tilde{\omega}_I \left(\Omega_{\text{eff}} + \hbar\tilde{\omega}_I\right)}{\left[\Delta_h^2 - 2\hbar\tilde{\omega}_I \left(\Omega_{\text{eff}} + 2\hbar\tilde{\omega}_I\right)\right]^2} + \mathcal{O}\left(\Lambda^3\right). \qquad (15)$$

The parametric dependence of this energy correction for $\Omega_{R0} = \omega_I$ and $Z_{\text{eff}} = 1$ is depicted in Fig. 2(a). It becomes evident that $\Delta E_0$ is relatively small except for the region of large effective masses and large attractive bath-impurity interactions, see the blue region in Fig. 2(a). For completeness, we note that from the functional form of Eq. (15) it is easy to predict that the amplitude of $\Delta E_0$ reduces for increasing $\Omega_{R0}$.

The minimum value of $|\langle \hat{S} \rangle|$, $|\langle \hat{S} \rangle|_{\min}$, within second order perturbation theory in $\Lambda$, occurs at $\Delta = -\frac{E_0}{2} + \frac{\Delta_h}{2} + \Delta E_0$, and is demonstrated in Fig. 2(b) for different $\Omega_{R0}$ and $Z_{\text{eff}}$. Apparently, $|\langle \hat{S} \rangle|_{\min} \to 1$ except for $m_I^* \to \infty$ where the suppression of $|\langle \hat{S} \rangle|_{\min}$ is significant. However, for increasing $\Omega_{R0}$ it turns out that $|\langle \hat{S} \rangle|_{\min} \to Z_{\text{eff}}/2$, see Fig. 2(c).

This result is in qualitative agreement with Fig. 1(d), supporting the fact that the light-induced shift of the effective potential is at least partly responsible for the observed increase of $|\langle \hat{\boldsymbol{S}} \rangle|_{\min}$. As we shall argue in the following section this outcome can be independently verified by comparing the predictions of the effective potential with the ML-MCTDHX calculations.

# 4 Comparison with many-body simulations: Competition of impurity dressing and impact of Rabi-coupling

To investigate the accuracy of the effective description we present in Fig. 3, the variance of the impurity position $\Delta x_I \equiv \sqrt{\langle \Psi_0 | \hat{x}_I^2 | \Psi_0 \rangle - \langle \Psi_0 | \hat{x}_I | \Psi_0 \rangle^2}$ and momentum $\Delta p_I \equiv \sqrt{\langle \Psi_0 | \hat{p}_I^2 | \Psi_0 \rangle - \langle \Psi_0 | \hat{p}_I | \Psi_0 \rangle^2}$ as a function of the detuning within the fully correlated ML-MCTDHX approach and the effective potential of Eq. (10) which is equivalent to Eq. (12), for $g_{BI} = -g_{BB} = -0.5\sqrt{\frac{\hbar^3 \omega_B}{m_B}}$ and $N_I = 1$. Additionally, in order to estimate the deviation of the impurity ground state from the Gaussian profile expected within the harmonic approximation we also provide $U_I = \Delta x_I \Delta p_I / \hbar - 1/2$. Recall that $U_I = 0$ for any squeezed coherent state, and thus $U_I > 0$ consists a quantitative estimator regarding deviations from an effectively harmonically trapped non-interacting impurity.

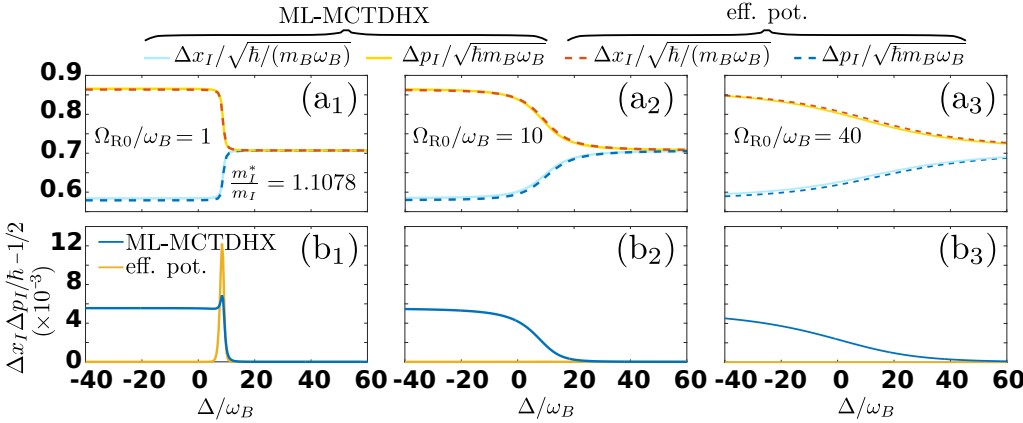

Figure 3: $(a_i)$ The momentum, $\Delta p_I$ and position, $\Delta x_I$ uncertainties for the impurity species with varying detuning, $\Delta$. The results are provided for different $\Omega_{R0}$ (see inset labels for $i = 1, 2, 3$) and within the effective potential and ML-MCTDHX approach (see legend). Excellent agreement between the two methods is observed. $(b_i)$ The deviation of the product $\Delta x_I \Delta p_I$ from the bound set by the Heisenberg uncertainty principle. In all cases, $g_{BI} = -g_{BB} = -0.5\sqrt{\hbar^3 \omega_B / m_B}$, $N_B = 100$ and $m_I = m_B$. Small deviations between the ML-MCTDHX and effective potential models occur for $\Delta < -E_{1\uparrow}/\hbar$ due to the correlated character of the polaronic state.

We find that using an effective mass of $m_I^* = 1.071$, the effective potential accurately captures the ML-MCTDHX behavior of both $\Delta x_I$ and and $\Delta p_I$ with respect to the detuning and different $\Omega_{R0}$, see Fig. 3$(a_i)$, with $i = 1, 2, 3$. In fact, this value of $m_I^*$ has been selected such that the width of the impurity density for $|\Psi_{B+1\uparrow}\rangle$ obtained within the correlated approach is reproduced by the effective potential model. In addition, we fix $Z_{\text{eff}} = 1$ throughout this section since the ML-MCTDHX data predict $Z = 0.984$ which is close to the effective potential result, i.e. $\langle \psi_{0\uparrow}^I | \psi_{0\downarrow}^I \rangle = 0.9893$ for $\Omega_{R0} = 0$. Figures 3$(a_i)$,

$i = 1, 2, 3$, showcase that the effective potential is adequate for correctly describing the impurity state, as it captures accurately both the position and momentum uncertainties. The most notable deviations among the two approaches is that the asymptotic values of $\Delta x_I$, $\Delta p_I$ for $\Delta \to -\infty$ within the effective potential approximation slightly deviate from the correlated approach.

To gain further insights into the discrepancies between the two approaches we next rely on the uncertaintly $U_I$ data, see Fig. 3($b_i$), with $i = 1, 2, 3$. It can be observed that for all employed values of $\Omega_{R0}$ the deviations of the impurity state within ML-MCTDHX from the minimal uncertainty limit are relatively small exhibiting a maximum of $U_I = 0.007$. This behavior indicates that the impurity distribution is close to a squeezed coherent state. Furthermore, the uncertainty $U_I$ features a saturation tendency for $\Delta \to -\infty$ which is attributed to the fact that in this limit the polaron state corresponding to spin-$\uparrow$ becomes the ground state of the system. In addition for $\Omega_{R0} = 1\omega_B$, see Fig. 3($b_1$), an uncertainty peak appears close to resonance at $\Delta \approx -E_{1\uparrow}/\hbar$. This peak is also captured by the effective potential which for larger $\Omega_{R0}$ yields $U_I < 10^{-6}$, irrespectively of $\Delta$ [Fig. 3($b_2$), ($b_3$)] implying an almost perfect harmonic oscillator state. The saturation of $\Delta x_I \Delta p_I$ to a value larger than $\hbar/2$ demonstrates that the phonon coupling of the polaron introduces a momentum uncertainty that cannot be solely accounted through a renormalization of the effective mass and trap of the polaron.

This behavior can be explained by considering that a polaron state exhibits a finite Tan's contact [153–155] associated with a $1/k^4$ tail of the momentum distribution [36] which naturally cannot be modeled by an effective potential model. Nevertheless, as it can be deduced from Fig. 3($b_1$), this effect leads to a small modification from an ideal harmonic confinement. In contrast, the peak at $\tilde{\Delta} = 0$ is traced back to the admixture of $| \downarrow \rangle$ and $| \uparrow \rangle$ states with different spatial distributions manifesting the non-negligible spin-orbit coupling ($\Lambda \neq 0$) in this parametric region, see also the related discussion in Sec. 3.3. Recall that by construction the effective potential takes this effect fully into account. The decrease of the $U_I$ peak amplitude in the correlated case can be attributed to the reduction of the overlap between the $| \downarrow \rangle$ and $| \uparrow \rangle$ spin configurations due to the $\Omega_{R0}$-dependent phononic dressing of the impurity which is not captured by the effective potential approach. For larger values of $\Omega_{R0}$ since the energy gap among the light-coupled spin-eigenstates $|+\rangle$ and $|-\rangle$ of $\hat{H}_S$ is given by $Z_{\text{eff}}\Omega_{R0} \gg \omega_B$ such effects become negligible, see also Eq. (15) and Appendix B.2.

| $\Omega_{R0}$ | $|\langle \hat{\boldsymbol{S}} \rangle|_{\min}$ ML-MCTDHX | $|\langle \hat{\boldsymbol{S}} \rangle|_{\min}$ effective potential | $Z_{\text{eff;fit}}$ |
|---|---|---|---|
| 1 | 0.493629 | 0.498029 | 0.991165 |
| 10 | 0.499246 | 0.499829 | 0.998835 |
| 40 | 0.499931 | 0.499985 | 0.999891 |

Table 1: Comparison between ML-MCTDHX obtained values of the minimum value of $|\langle \hat{\boldsymbol{S}} \rangle|$, see Fig. 1(d), with the effective potential approach. For the effective potential approach we use the second order perturbative results for $\Lambda$, see Appendix B.2, and we fix the parameters to $m_I^* = 1.1078$ and $Z_{\text{eff}} = 1$. The minimum within this approach is assumed to be at $\Delta = -\frac{E_0}{2} + \frac{\Delta_h}{2} + \Delta E_0$. To demonstrate the accuracy of the effective potential results we underline the digits of the ML-MCTDHX results that are captured correctly by this approximation. $Z_{\text{eff;fit}}$ is the value of $Z_{\text{eff}}$ such that the effective potential $|\langle \hat{\boldsymbol{S}} \rangle|_{\min}$ matches the ML-MCTDHX one.

Turning back to the open issue of the reduction of $|\langle \hat{\boldsymbol{S}} \rangle|$ for higher $\Omega_{R0}$ observed in Fig. 1(d), we show in Table 1 that the most important contribution to the increase of

$|\langle \hat{\boldsymbol{S}} \rangle|$ is the change of the modification of the effective potential for larger $\Omega_{\text{R0}}$. However, Fig. 3($b_i$), $i = 1, 2, 3$ provide an indication that the polaron dressing also competes with the light-dressing as $\Omega_{\text{R0}}$ increases. Indeed, stronger dressing leads to lower $U_I$, as the transition from the polaronic value for $\Delta \to -\infty$ to the non-interacting value of $U_I = 0$ for $\Delta \to \infty$ becomes less sharp, compare Fig. 3($a_1$) to Fig. 3($a_2$) and ($a_3$). This according to our discussion above implies a smaller Tan's contact and thus effectively smaller impurity-bath interactions. This is also supported by $Z_{\text{eff;fit}}$, see Table 1, being the value of $Z_{\text{eff}}$ such that the effective potential prediction for $|\langle \hat{\boldsymbol{S}} \rangle|_{\text{min}}$ reproduces the ML-MCTDHX one. It can be clearly observed that the value of $Z_{\text{eff}}$ increases with $\Omega_{\text{R0}}$ showing that the polaron tends to undress from its phononic cloud as the light-spin coupling increases.

Therefore, we conclude that the effective potential is able to adequately characterize the system for $g_{BI} < g_{BB}$. Also, the effective mass of the impurity is unambiguously identifiable by studying the dependence of the uncertainties $\Delta x_I$ and $\Delta p_I$ on $\Delta$. This motivates the investigation on whether the effective potential enables the characterization of the system for $g_{BI} \geq g_{BB}$.

# 5   Repulsive Bose polaron

The repulsive Bose polaron, unlike its attractive counterpart, is stable only in the case that $g_{BI}$ is sufficiently small such that phase-separation among the bath and the impurity species is prevented [69, 156]. This yields three different interaction regimes (dictated by the miscibility condition of the impurity with its environment) for studying the compliance of the Rabi-coupled impurity with the effective potential description. Here, we will focus on the most interesting of these regimes. Namely, the case of immiscible bath-impurity interactions $g_{BI} > g_{BB}$ and close to the transition point $g_{BI} = g_{BB}$. Indeed, our previous studies, see Ref. [69–72], demonstrate that deep in the miscible regime $g_{BI} < g_{BB}$ a similar behavior to the attractive case occurs, which we also have independently verified for the current setup (not shown for brevity).

## 5.1   Impurity light-dressing at the phase-separation threshold

Before analyzing the repulsive Bose polaron in the interaction regime $g_{BI} > g_{BB}$, let us briefly discuss its properties for $g_{BI} = g_{BB}$. In the absence of light dressing of the impurities, i.e. $\Omega_{\text{R0}} = 0$, it is well established that the effective potential of the impurity is approximately a box potential, see Fig. 4(a). Focusing on $\Omega_{\text{R0}} \neq 0$, however, two different behaviors are encountered: i) $\hbar\Delta \ll -E_{1\uparrow}$ reproduces the same effective potential properties, but ii) for $\hbar\Delta \approx -E_{1\uparrow}$ the system's characteristics alter prominently as we elaborate below.

To elucidate the back-action of the light-dressed polaron, we present in Fig. 5($a_1$) the modification of the spatially resolved bath density, $\delta\rho_B^{(1)}(x) = \rho_B^{(1)}(x) - \rho_{B0}^{(1)}(x)$, where $\rho_B^{(1)}(x) = \langle\Psi_0|\hat{\Psi}_B^\dagger(x)\hat{\Psi}_B(x)|\Psi_0\rangle$ is the density of the bath species. Recall that $|\Psi_0\rangle$ is the ML-MCTDHX calculated interacting ground state wavefunction, while the density of the bath in the absence of the impurity, i.e. $g_{BI} = 0$, is $\rho_{B0}^{(1)}(x)$. In addition, we show the spatially resolved impurity spin densities along the $x$ [Fig. 5($b_1$)] and $z$ [Fig. 5($b_1$)] axes which are computed as

$$\langle\hat{S}_\mu(x)\rangle = \frac{1}{2} \sum_{\alpha,\beta\in\{\uparrow,\downarrow\}} \sigma_{\alpha\beta}^\mu \langle\Psi|\hat{\Psi}_\alpha^\dagger(x)\hat{\Psi}_\beta(x)|\Psi\rangle. \tag{16}$$

These quantities allow us to study the modification of the polarization of the impurity

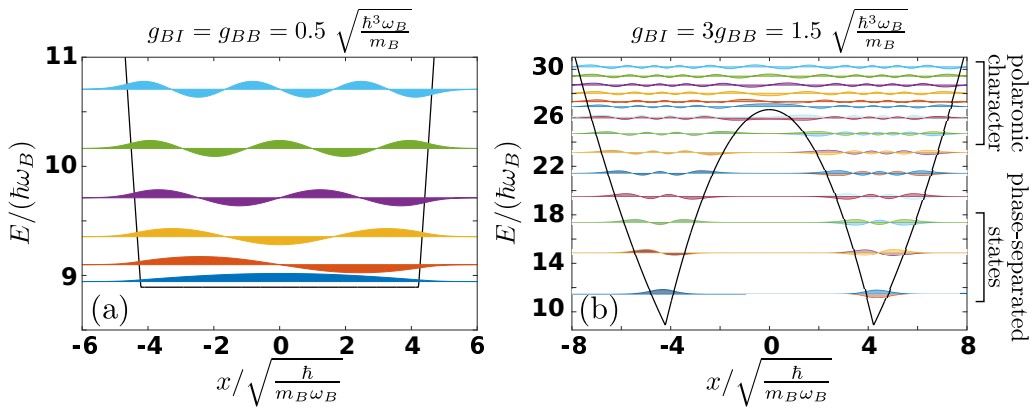

Figure 4: The effective potential, $V_{\text{eff}}(x) = \frac{1}{2}m_I\omega_I^2 x^2 + g_{BI}\rho_B^{(1)}(x)$, for $\Omega_{\text{R0}} = 0$, alongside its single-particle eigenstates. Here, we assume a Thomas-Fermi profile, Eq. (11). $V_{\text{eff}}(x)$ at (a) the critical interaction strength for phase-separation $g_{BI} = g_{BB} = 0.5 \sqrt{\frac{\hbar^3\omega_B}{m_B}}$ and (b) within the temporal orthogonality catastrophe regime $g_{BI} = 3g_{BB} = 1.5 \sqrt{\frac{\hbar^3\omega_B}{m_B}}$. In panel (b) we schematically assign the stable phase-separated eigenstates and the metastable polaronic ones [157]. For more details on the $\Omega_{\text{R0}} = 0$ effective potential, see Ref. [69].

spin-state in a spatially-resolved manner. Positive (negative) values of $\langle \hat{S}_\mu(x) \rangle$ indicate preferential occupation of the spin-↑ (spin-↓) state along the $\mu \in \{x, z\}$ spin-axis.

Notice, that the Hamiltonian of Eq. (1) does not contain any term proportional to $\hat{S}_y(x)$, implying that $\langle \hat{S}_y(x) \rangle = 0$. Consequently, the spin polarization of the impurity lies entirely along the $x$–$z$ plane, and hence the spatially resolved spin components $\langle \hat{S}_x(x) \rangle$ and $\langle \hat{S}_z(x) \rangle$ suffice to fully describe its local spin orientation. Notably, the local spin-density matrix can be expressed as

$$\rho_{\alpha\beta}^{(1)}(x) = \frac{1}{2}\rho_I^{(1)}(x) + \langle \hat{S}_x(x) \rangle \sigma_{\alpha\beta}^x + \langle \hat{S}_z(x) \rangle \sigma_{\alpha\beta}^z \qquad (17)$$

with $\alpha, \beta \in \{\uparrow, \downarrow\}$ and $\rho_I^{(1)}(x) = \sum_{\alpha \in \{\uparrow,\downarrow\}} \langle \Psi_0 | \Psi_\alpha^\dagger(x)\Psi_\alpha(x) | \Psi_0 \rangle$ is the spin-unresolved impurity density. As such, the three observables $\rho_I^{(1)}(x)$, $\langle \hat{S}_x(x) \rangle$ and $\langle \hat{S}_z(x) \rangle$ will enable us to fully appreciate the rise of local spin-correlations in the system.

For $\hbar\Delta \gg E_{1\uparrow} \approx -8.43~\hbar\omega_B$ the impurity predominantly lies in its non-interacting spin-↓ state since $\langle \hat{S}_z(x) \rangle \approx -\frac{1}{2}\rho_I^{(1)}(x) < 0$, see Fig. 5(c$_1$) and 5(d$_1$) for $\Delta \approx 40\omega_B$. However, the small population of the $|\uparrow\rangle$ polaronic state especially as $\hbar\Delta \approx E_{1\uparrow}$ is approached indicates a weak light-dressing of the impurity state associated with $\langle \hat{S}_x(x) \rangle < 0$ in Fig. 5(b$_1$) for $0 \leq \Delta/\omega_B < 20$. In contrast, for $\hbar\Delta \ll E_{1\uparrow}$ the polaron $|\uparrow\rangle$ state is almost perfectly reproduced, $\langle \hat{S}_z(x) \rangle \approx \frac{1}{2}\rho_I^{(1)}(x) > 0$, see Fig. 5(c$_1$) and 5(d$_1$) for $\Delta \approx -60\omega_B$. Due to the box-like effective potential [Fig. 4(a)] the density of the impurity is only constrained by the Thomas-Fermi radius of the BEC, see Fig. 5(c$_1$). Indeed, as Fig. 4(a) shows the bath-impurity interaction almost perfectly counteracts the harmonic trapping potential of the impurity, in the spatial regime where it has a finite density, $|x| \leq R_{\text{TF}}$. The bath reacts to the presence of the impurity by creating a density dip at the location of the impurity and by expelling a small part of its density from the center of the trap, see Fig. 5(a$_1$) for $\Delta < -10\omega_B$. This is an explicit imprint of the repulsive interaction to the spin-↑ impurity component. Notice that $\langle \hat{S}_x(x) \rangle$ tends to 0 much more rapidly as $\Delta$ decreases for $\Delta < -E_{1\uparrow}/\hbar$ when compared to the case that $\Delta$ increases for $\Delta > -E_{1\uparrow}/\hbar$,

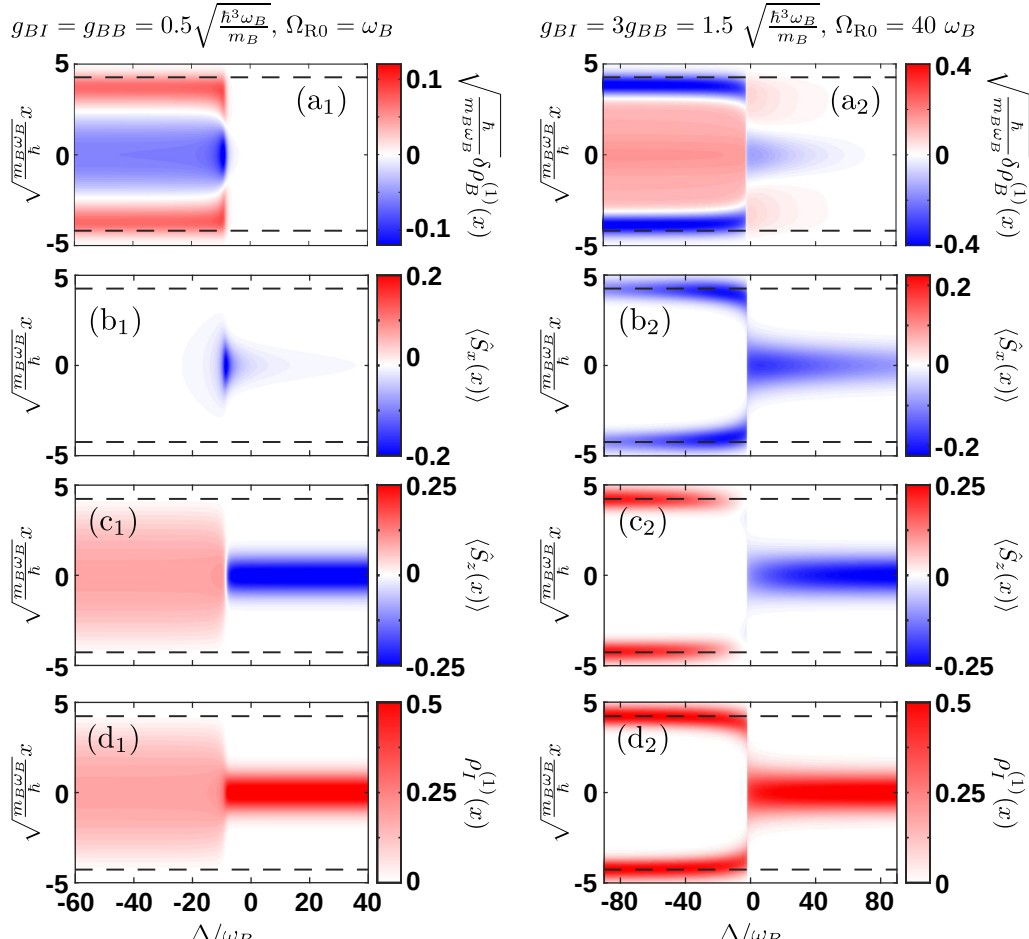

Figure 5: Behavior of the bath and impurity densities in the repulsive polaron case. ($a_i$) Density fluctuations of the BEC background, $\delta\rho_B^{(1)}(x)$ with respect to $\Delta/\omega_B$. Spatially resolved expectation values of the spin operators ($b_i$) $\hat{S}_x(x)$ and ($c_i$) $\hat{S}_z(x)$ for varying $\Delta$. ($d_i$) The total (spin-unresolved) impurity density $\rho_I^{(1)}(x)$ in terms of $\Delta$. In all cases, the index $i = 1$ refers to $g_{BI} = g_{BB} = 0.5\sqrt{\hbar^3\omega_B/m_B}$ and $\Omega_{R0} = \omega_B$, whilst $i = 2$ corresponds to $g_{BI} = 3g_{BB} = 1.5\sqrt{\hbar^3\omega_B/m_B}$ and $\Omega_{R0} = 40\,\omega_B$. Superposition states of the impurity manifest by the $\langle\hat{S}_x(x)\rangle < 0$ regions in ($b_i$). The dashed lines mark the Thomas-Fermi radius of the BEC.

see Fig. 5($b_1$). This is because the ground state involving the interacting spin-↑ impurity (i.e. the polaron state, $\Delta \to -\infty$) has small overlap with the non-interacting ground state with spin-↓ (reproduced for $\Delta \to \infty$) and thus the light-dressing of the impurity is hindered by the small spatial overlap of the Rabi-coupled states, see also Fig. 5($c_1$).

In the case of $\Delta \approx -E_{1\uparrow} \approx -8.43\,\hbar\omega_B$ the impurity is predominantly found in a light-dressed superposition state of spin-↑ and spin-↓ where $\langle\hat{S}_x(x)\rangle \approx -\frac{1}{2}\rho_I^{(1)}(x) < 0$, see Fig. 5($b_1$). The size of the impurity ground state density is consistent with the one obtained using a renormalized trap frequency $\bar{\omega}_I \approx \frac{1}{2}\omega_I$, see also Eq. (14c). Notice, however, here that the value of $\bar{\omega}_I$ can be further refined by considering the mass renormalization due to polaronic dressing $m_I^* > m_I$. Nevertheless, it can also be seen that the spin density of the impurity is modified throughout its spatial extent as evident by the curved node $\langle\hat{S}_z(x)\rangle = 0$ in the $\Delta$–$x$ plane observed in this $\Delta$ regime, see Fig. 5($c_1$) for $-12 < \Delta/\omega_B < -5$. This implies a slightly different density between the spin-↓ and spin-↑ impurity states

which are related with a squeezing operation, generalizing the spin-orbit coupling term $\sim \Lambda \left[\left(\hat{a}^{\dagger}\right)^{2} + \hat{a}^{2}\right] \hat{\sigma}_{z}$ of the effective potential Hamiltonian of Eq. (12). Indeed, it can be verified that the effective potential Hamiltonian of Eq. (10) describes the behavior of the system in this regime producing almost indistinguishable $\langle \hat{S}_{x}(x) \rangle$ and $\langle \hat{S}_{z}(x) \rangle$ to Fig. 5($b_1$). Similarly to the $g_{BI} < g_{BB}$ scenario the impact of the state squeezing for distinct spin-components becomes negligible for increasing $\Omega_{R0} \geq 10\omega_{B}$.

The above observations explicate that even in the case of $g_{BI} = g_{BB}$, the qualitative description of the system is similar in the weakly attractive and repulsive interaction regimes. Of course, on a quantitative level the modeling of the system is slightly more complicated since the spatial extent $|x| \geq R_{TF}$ should also be taken into account. Thus, the simplified version of the effective potential given by Eq. (12) assuming the TF approximation is not valid, necessitating a numerical treatment of the general effective model described by Eq. (10).

## 5.2 Impurity light-dressing in the phase-separation regime

When phase-separation occurs, $g_{BI} > g_{BB}$, the behavior of the system becomes drastically different as compared to the miscible region, $g_{BI} \leq g_{BB}$. Here, the ensuing low-lying in terms of energy states of an interacting impurity within the effective potential predominantly reside outside the $R_{TF}$ of the BEC, see Fig. 4(b) and especially the states marked as phase separated. In fact, these states have almost zero overlap with the ground state of the non-interacting system, a phenomenon that has been understood as the origin of the temporal orthogonality catastrophe [69]. Within this framework the states of polaronic character correspond to highly-excited states of the effective potential near the top of the effective barrier at $x \approx 0$, see Fig. 4(b), which are shown to be dynamically unstable due to their beyond effective-potential coupling to the bath. We remark that a similar phase-separation phenomenology was also reported in the three-dimensional low temperature case [156].

In the case of light-impurity dressing the above property of the phase separated system, characterized here by $g_{BI} = 3g_{BB} = 1.5 \sqrt{\frac{\hbar^{3}\omega_{B}}{m_{B}}} > g_{BB}$, leads to extremely sharp transitions (less that $\delta\Delta = 0.01\omega_{B}$ wide) between the non-interacting spin-$\downarrow$ ground state for $\Delta > -E_{1p} = -11.2\omega_{B}$ and the interacting spin-$\uparrow$ phase separated state for $\Delta < -E_{1p}$ for small $\Omega_{R0}$. This is exemplarily illustrated in Fig. 6(a) for $\Omega_{R0} = \omega_{B}$. As $\Omega_{R0}$ increases, the excited states of the interacting spin-$\uparrow$ impurity get involved in the light-matter dressing resulting in a positive shift of the resonance which tends to $\Delta = 0$ for large $\Omega_{R0}$. This is accompanied by a noticeable dressing of the different spin-states close to the transition. Nevertheless, these transitions remain somewhat sharp with widths of the order of $\delta\Delta \sim 0.1\omega_{B}$ and $\delta\Delta \sim 1\omega_{B}$ for $\Omega_{R0} = 10\omega_{B}$ and $\Omega_{R0} = 40\omega_{B}$ respectively, the latter is better visible in Fig. 6(b).

To reveal how the impurity behaves in this interaction regime, we provide the impurity and bath density modifications in Fig. 5($a_2$), ($b_2$), ($c_2$) and ($d_2$), for large $\Omega_{R0} = 40\omega_{B}$. As in all studied cases, the response of the bath and the impurity density for $\Delta \gg -E_{1p}$ and $\Delta \ll -E_{1p}$ are approximately the same as the non-interacting and the interacting ground state of the impurity respectively. This is best appreciated by comparing Fig. 5($c_2$) and 5($d_2$) for $\Delta \approx \pm 80\omega_{B}$. However, the effects of the light-dressing of the impurity are already obvious for quite sizable detunings e.g. for $\Delta > -3\omega_{B}$ in Fig. 5($a_2$), where a dramatic change of the impurity state occurs. Notice, that the state of the impurity is quite different for more negative detunings than $\Delta \approx -3\omega_{B}$ when compared to detunings towards positive values. Indeed, in the latter case, we observe that the impurity predominantly resides around the trap center (i.e. within the BEC), as both spin-components $\langle \hat{S}_{x}(x) \rangle$

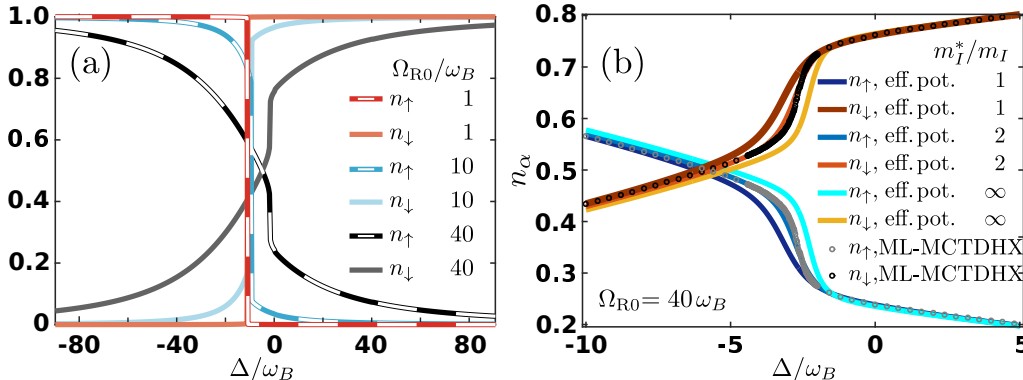

Figure 6: (a) Population of the spin-↑ and spin-↓ impurity states with respect to the detuning $\Delta$ at different $\Omega_{R0}$ (see legend) and within the strongly interacting case $g_{BI} = 3g_{BB} = 1.5\sqrt{\hbar^3\omega_B/m_B}$. The sharp transition for small $\Omega_{R0}$ becomes gradually smoother and shifts as $\Omega_{R0}$ increases. (b) Comparison of the ML-MCTDHX data for $\Omega_{R0} = 40\omega_B$ with the effective potential for $Z_{eff} = 1$ and varying $m_I^*$ (i.e. without fitting parameters). The excellent agreement of the corresponding data leads to the conclusion that the effective mass of the polaron within ML-MCTDHX is $m_I^* = 2m_I$. Notice that the range of $\Delta$ values is restricted such that the deviations of $n_\alpha$ between different values of $m_I^*$ are better visible.

and $\langle \hat{S}_z(x) \rangle$ indicate, see Fig. 5(b$_2$) and (c$_2$) respectively. Here, energetically higher-lying interacting states corresponding to the ones on top of the barrier of the double-well effective potential, see the eigenstates marked by the top bracket in Fig. 4(b), are involved. In the opposite scenario of $\Delta < -3\omega_B$, these eigenstates do not participate. Rather, the impurity is found at the minima of its double well effective potential, located at the edges of the spatial extent of the Thomas-Fermi radius, see Fig. 5(b$_2$) and (c$_2$). Therefore, the highly asymmetric form of the populations of the spin-↑ and spin-↓ states with respect to their crossing point, observed in Fig. 6(a) for $\Omega_{R0} = 10\ \omega_B$ and $\Omega_{R0} = 40\ \omega_B$, can be attributed to the different nature of the involved interacting states. Finally, let us note that the bosonic host responds to the population of the spin-↑ states by a small expulsion of bosonic density from the location of the impurity, see Fig. 5(a$_2$), see the blue parts of Fig. 5(a$_2$) in comparison to Fig. 5(c$_2$).

This analysis implies that the parametric region of detunings $\Delta \leq -3\omega_B$ is especially suited for studying the strongly repulsive Bose polaron in one-dimension as it allows to explore strongly interacting polarons by counteracting their decay mechanism enforced by the temporal orthogonality catastrophe effect [69]. Therefore, an important open question is which properties of the strongly repulsive Bose polaron can be examined in this manner. The effective-potential model of Eq. (10) reveals that the population of the spin-↑ and spin-↓ states in the region where the dressing changes depends sensitively on the effective mass of the polaron, see Fig. 6(b). Here, we have set $Z_{eff} = 1$, i.e. we assume that the light-dressing of the impurity is so strong as to couple to the light-dressed state to the phononic bath, thus not affecting the overlap of the spin-↑ and spin-↓ state. When comparing to the ab-initio ML-MCTDHX data we observe that they fit almost perfectly to the effective-potential curves for $m_I^* = 2m_I$. Thus, detailed comparisons of experimentally obtained polaron data with the predictions of the effective potential of Eq. (10) (or its higher-dimensional generalizations including also relevant bound state channels) might be relevant in order to experimentally identify the polaron effective mass in the strongly interacting regime.

# 6 Conclusions and perspectives

By carefully analyzing *ab-initio* simulations with suitably constructed effective models we have demonstrated the validity of an effective potential approach for capturing the state of simultaneous light and phonon dressed spinor impurities. Specifically, only one spin-state interacts with the structureless bosonic host with the other being uncoupled. The considered impurity-bath interactions are either attractive or repulsive and hence both attractive and repulsive Bose polaron properties, such as the energy, residue and effective mass are assessed. Our results are in line with our previous studies on the dynamical properties of Bose polarons [70, 71, 81, 82] utilizing a variational many-body method. However, they go beyond them by means of facilitating the experimental identification, within a relatively simple and efficient framework, of polaronic properties that are tricky to unambiguously determine in trapped Bose gases [68, 70].

Starting from a two-level model we systematically build up an effective potential which contains i) a renormalized spin-dependent trap frequency due to the bosonic host manifesting as a state-dependent detuning, ii) a modified trap length and iii) a spin-orbit coupling term. The trap length and frequency shifts are regulated by the effective polaron mass. It is showcased that this effective potential can, at least qualitatively, describe the Bose polarons experiencing light dressing. In particular, it corroborates our numerical studies indicating the competing character of the phononic and light dressing. This showcases the tendency of the impurity to decouple from the fluctuations of its host as the light-impurity coupling increases. An additional highlight is the stabilization of the strongly repulsive Bose polaron against temporal orthogonality catastrophe, which allows the investigation of strongly interacting polaron physics without the need of complex experimental spectroscopic schemes [71].

There are several promising avenues for future research that build upon the findings of this study. It would be advantageous to extend the current findings to higher spatial dimensions. One important consideration in these settings is whether the formation of Efimov states substantially modifies the properties and formation dynamics of the Bose polarons [158–163]. Furthermore, it is essential to examine the robustness of the ground state properties of the system in the current and higher-dimensional settings, particularly at finite temperatures [156]. The case of strong attraction is also a promising avenue for elucidating the influence of strong polaron-polaron interactions and their bound states on the light dressing of impurities.

## Acknowledgements

**Funding information** G.M.K. has received funding by the Austrian Science Fund (FWF) [DOI: 10.55776/F1004]. S.I.M acknowledges support from the Missouri University of Science and Technology, Department of Physics, Startup fund. F.G. acknowledges funding by the Deutsche Forschungsgemeinschaft (DFG, German Research Foundation) under Germany's Excellence Strategy – EXC-2111 — 390814868. H.R.S. acknowledges support for ITAMP by the NSF. P.S. acknowledges funding by the Cluster of Excellence "Advanced Imaging of Matter" of the Deutsche Forschungsgemeinschaft (DFG) - EXC 2056 - project ID 390715994.

# A  Many-body numerical approach

To address the ground state properties of the Rabi coupled spin-1/2 impurities embedded in an one-dimensional bosonic environment we rely on the multilayer multiconfiguration time-dependent Hartree method for atomic mixtures (ML-MCTDHX) [145–147]. ML-MCTDHX is an *ab-initio* variational method which employs a time-dependent and variationally optimized basis for representing the many-body wavefunction. In particular, ML-MCTDHX features a multi-layered ansatz allowing for the variational optimization of both the single-particle and species time-dependent bases. This facilitates capturing the many-body Hilbert space and as a consequence the involved correlation properties of the system.

Specifically, to account for interspecies correlations the many-body wavefunction is initially expanded in terms of $D$ distinct species functions, $|\Psi_k^\sigma(t)\rangle$, $i = 1, \ldots, D$, for the bath $\sigma = B$ and the impurity $\sigma = I$. Hence, we arrive in a so-called truncated Schmidt decomposition of order $D$

$$|\Psi(t)\rangle = \sum_{k=1}^{D} \sqrt{\lambda_k} |\Psi_k^B(t)\rangle \otimes |\Psi_k^I(t)\rangle, \tag{18}$$

with expansion (Schmidt) coefficients $\lambda_k$. This expansion has explicit physical implications by means that entanglement among the species is present if two or more $\lambda_k$'s possess non zero values adhering to the interspecies correlations emanating in the system. Otherwise, Eq. (18) has a tensor product form and entanglement is absent since $\lambda_1 = 1$ and $\lambda_k = 0$, for $k = 2, \ldots, D$.

To properly account for the intraspecies correlations of the multicomponent system each $|\Psi_k^\sigma(t)\rangle$, $i = 1, \ldots, D$, is further expressed with respect to a time-dependent number-state basis

$$|\Psi_k^\sigma(t)\rangle = \sum_{\vec{n}} A_{k;\vec{n}}^\sigma(t) |\vec{n}(t)\rangle^\sigma, \tag{19}$$

where $A_{k;\vec{n}}^\sigma(t)$ refer to the expansion coefficients. Also, $\vec{n} = (n_1, \ldots, n_{M_\sigma})$ is the vector of particle occupations of each of the $M_\sigma$ distinct time-dependent single-particle functions, $|\phi_j^\sigma(t)\rangle$, $j = 1, \ldots, M_\sigma$, that satisfy $\sum_{j=1}^M n_j = N_\sigma$. Finally, the above-mentioned single-particle functions are expanded in a time-independent single-particle basis, $\chi_l(x)$. This expansion for the bath species reads

$$|\phi_j^B(t)\rangle = \sum_{k=1}^{\mathcal{M}} \phi_{j;l}^B(t) \int \mathrm{d}x \; \chi_l(x) \hat{\Psi}_B^\dagger(x) |0\rangle. \tag{20}$$

On the other hand, for the impurity it explicitly takes into account the spin-1/2 degree of freedom

$$|\phi_j^I(t)\rangle = \left[ \sum_{k=1}^{\mathcal{M}} \int \mathrm{d}x \; \phi_{j;l\uparrow}^I(t) \chi_l(x) \hat{\Psi}_\uparrow^\dagger(x) + \phi_{j;l\downarrow}^I(t) \chi_l(x) \hat{\Psi}_\downarrow^\dagger(x) \right] |0\rangle. \tag{21}$$

Accordingly, the time-evolution of the many-body wavefunction is determined by the expansion coefficients $\lambda_k(t)$, $A_{k,\vec{n}}^\sigma(t)$ and $\phi_{j;l}^\sigma(t)$, which can be obtained by solving the ML-MCTDHX equations of motion. These are numerically computed through a variational principle such as the Dirac-Frenkel one [164,165] and utilizing the wavefunction expansion explicated in Eqs. (18), (19), (20) and (21). This procedure results in $D^2$ coupled linear equations for the Schmidt coefficients, $\lambda_k(t)$, in addition to $D\binom{N_B+M_B-1}{M_B-1} + D\binom{N_B+M_B-1}{M_B-1}$ and $M_B + M_I$ non-linear integrodifferential equations for the species functions and single-particle functions respectively. To testify the reliability of the ML-MCTDHX results we

increase the number of Schmidt coefficients and impurity single-particle functions up to $D = M_I = 12$ and the number of bath single-particle functions up to $M_B = 4$ observing the convergence of the observables of interest.

In this study, we evaluate the ground state properties of the many-body Hamiltonian of Eq. (1) via the so-called improved relaxation scheme implemented within the ML-MCTDHX framework. Improved relaxation is an iterative scheme that is used to optimize the many-body basis referring to the $A^\sigma_{k,\vec{n}}(t)$ and $\phi^\sigma_{j;l}(t)$ coefficients for the variationally optimal representation of the ground state. This scheme is initialized with an arbitrary initial many-body basis and subsequently for each step of the iteration the total energy of the system is minimized by evaluating the lowest in energy eigenvector within the basis spanned by the species functions, $|\Psi^\sigma_k\rangle$, followed by the imaginary time propagation of the $A^\sigma_{k,\vec{n}}(t)$ and $\phi^\sigma_{j;l}(t)$ coefficients in a fixed time-interval. Each iterative step results in the reduction of the energy expectation value, and hence the overall ground state of Eq. (1) is identified by the saturation of the energy of the many-body wavefunction to a prescribed accuracy, here, $\leq 10^{-12}$.

Finally, let us argue on the suitability of the ML-MCTDHX ansatz of Eqs. (18), (19), (20) and (21) for exploring the properties of few impurities embedded in a BEC. Recall that a Bose gas corresponds to a perfect BEC if only one single-particle state is occupied by all constituting particles, which implies that the many-body BEC state is described exactly for $M_B = 1$. In practice, away from the thermodynamic limit, $N \to \infty$, the BEC is slightly depleted. For small intraspecies interactions $g_{BB} < 1$ and moderate particle numbers $N_B = 100$ this depletion is suppressed. Therefore, only a small number of $M_B$ ensures the numerical convergence of such states. Strikingly, it has been shown that even the dynamics of a Bose gas proximal to a BEC state can be accurately explored by involving only a small number of single-particle states [69–72]. In addition, to the above it is well-known that the quasi-particle states such as polarons are characterized by a large overlap with the ground state of the system involving non-interacting impurities with its environment. Therefore, the expected entanglement among the impurities and the bath is rather small, implying that only a Schmidt decomposition of lower order [Eq. (18)] suffice for the accurate representation of such quasi-particle states. Therefore, the study of the expected physical properties of the Bose polaron problem motivate a truncation scheme in terms of the single-particle and single-species basis states which as previously mentioned lies at the heart of the ML-MCTDHX framework.

# B    Details on the effective potential Hamiltonian

## B.1    Justification of perturbation theory in $\Lambda$

The reason for the negligible corrections stemming from the effective potential of Eq. (12) is the small value of $\Lambda$. Notice that according to Eq. (12) the leading order correction to the two-level model is the coupling between the the spin-ground state of the spatial vaccum state $\hat{a}|0\rangle = 0$ and the second excited states $|2\rangle = \frac{(\hat{a}^\dagger)^2}{\sqrt{2}}|0\rangle$ of either spin. Therefore, this correction is of second order in perturbation theory $\propto \Lambda^2/4$ and the energy difference of the coupled states is at least $2\hbar\tilde{\omega}_I$. More specifically, the order of magnitude of this excitation relative to the characteristic energy scale of the system is

$$\frac{1}{\hbar\tilde{\omega}_I} \frac{\hbar^2\Lambda^2}{4 \times (2\hbar\tilde{\omega}_I)} = \frac{1}{8} \frac{\left(\frac{m_I}{m_I^*} + \frac{g_{BI}}{g_{BB}} \frac{m_I\omega_I^2}{m_B\omega_B^2} - 1\right)^2}{\left(1 + \frac{m_I}{m_I^*}\right)^2 \left(\frac{g_{BI}}{g_{BB}} \frac{m_I\omega_I^2}{m_B\omega_B^2} - 2\right)^2} \leq \frac{1}{8}\left(1 + \frac{m_I}{m_I^*}\right)^{-2}. \tag{22}$$

Here, we have used that i) $\frac{g_{BI}}{g_{BB}}\frac{m_I\omega_I^2}{m_B\omega_B^2} \leq 1$ so that the temporal orthogonality catastrophe is prevented [69] and the impurity is inside the BEC and ii) $0 \leq \frac{m_I}{m_I^*} \leq 1$ since the effective mass of the polaron is expected to be higher than the bare mass of the impurity. Therefore, the spin-orbit coupling term $\propto \Lambda$ in Eq. (12) can be treated as a perturbation since its coupling is at least one order of magnitude smaller than the characteristic energy scale.

## B.2  Second order perturbation theory in $\Lambda$

For $\Lambda = 0$, all the terms in the Hamiltonian of Eq. (12) are diagonal to $\hat{a}^\dagger\hat{a}$ and thus we can work in the number state basis, $|n\rangle = \frac{(\hat{a}^\dagger)^n}{\sqrt{n!}}|0\rangle$. In this case, the Hamiltonian reads

$$\hat{H}_{\mathrm{eff}} = \tilde{E}_0(n) + \frac{\hbar\tilde{\Delta}(n)}{2}\hat{\sigma}_z + \frac{\hbar\Omega_{\mathrm{eff}}}{2}\hat{\sigma}_x, \tag{23}$$

with $\tilde{E}_0(n) = \frac{E_0}{2} + \hbar\bar{\omega}_I(n+1/2)$ and $\tilde{\Delta}(n) = \Delta + E_0/\hbar - \Delta_h(n+1/2)$. This simplification yields the eigenenergies

$$\tilde{E}_\pm(n) = \tilde{E}_0(n) \pm \frac{\hbar}{2}\sqrt{\tilde{\Delta}^2(n) + \Omega_{\mathrm{eff}}^2}. \tag{24}$$

and eigenstates

$$|\Phi_{n,+}\rangle = |n\rangle \otimes \frac{\left(\tilde{\Delta}(n) + \sqrt{\tilde{\Delta}^2(n) + \Omega_{\mathrm{eff}}^2}\right)|\uparrow\rangle + \Omega_{\mathrm{eff}}|\downarrow\rangle}{\sqrt{\Omega_{\mathrm{eff}}^2 + (\tilde{\Delta}(n) + \sqrt{\tilde{\Delta}^2(n) + \Omega_{\mathrm{eff}}^2})^2}} \tag{25a}$$

$$|\Phi_{n,-}\rangle = |n\rangle \otimes \frac{-\Omega_{\mathrm{eff}}|\uparrow\rangle + \left(\tilde{\Delta}(n) + \sqrt{\tilde{\Delta}^2(n) + \Omega_{\mathrm{eff}}^2}\right)|\downarrow\rangle}{\sqrt{\Omega_{\mathrm{eff}}^2 + (\tilde{\Delta}(n) + \sqrt{\tilde{\Delta}^2(n) + \Omega_{\mathrm{eff}}^2})^2}} \tag{25b}$$

Second order perturbation theory reveals that a finite $\Lambda$ leads to the shift of the resonance for varying $\Omega_{\mathrm{eff}}$, namely

$$\delta E_0 = E_-^{(2)}(0) = -\frac{\hbar^2\Lambda^2}{2}\left[\frac{|\langle\Phi_{2,-}|\hat{\sigma}_z|\Phi_{0,-}\rangle|^2}{2\hbar\bar{\omega}_I} \right.$$
$$\left. + \frac{1 - |\langle\Phi_{2,-}|\hat{\sigma}_z|\Phi_{0,-}\rangle|^2}{2\hbar\bar{\omega}_I + \frac{1}{2}\sqrt{\tilde{\Delta}(0)^2 + \Omega_{\mathrm{eff}}^2} + \frac{1}{2}\sqrt{\tilde{\Delta}(2)^2 + \Omega_{\mathrm{eff}}^2}}\right] \tag{26}$$

which with some additional algebraic manipulations reduces to Eq. (15). In addition to this shift, we can show that the minimal value of $|\langle\hat{\boldsymbol{S}}\rangle|$ at resonance increases towards $Z_{\mathrm{eff}}/2$ for larger $\Omega_{\mathrm{R0}}$. Unfortunately, we cannot find a simple analytical expression for the value of $|\langle\hat{\boldsymbol{S}}\rangle|$, however, it can be straightforwardly calculated by the following expressions

$$|\langle\hat{\boldsymbol{S}}\rangle| = \sqrt{Z_{\mathrm{eff}}^2(\langle\hat{S}_x\rangle^2 + \langle\hat{S}_y\rangle^2) + \langle\hat{S}_z\rangle^2}, \tag{27}$$

with the individual components calculated via

$$\langle \hat{S}_x \rangle^2 = \left( 1 - \frac{\hbar^2 \Lambda^2}{4} \mathcal{M}_{0,-} \right) \langle \Phi_{0,-} | \hat{\sigma}_x | \Phi_{0,-} \rangle + \frac{\hbar^2 \Lambda^2}{2} \mathcal{M}_{0,+} \langle \Phi_{0,+} | \hat{\sigma}_x | \Phi_{0,-} \rangle$$

$$+ \frac{\hbar^2 \Lambda^2}{4} \left( \mathcal{M}_{2,+}^2 \langle \Phi_{0,+} | \hat{\sigma}_x | \Phi_{0,+} \rangle + \mathcal{M}_{2,-}^2 \langle \Phi_{0,-} | \hat{\sigma}_x | \Phi_{0,-} \rangle \right) \tag{28a}$$

$$+ \frac{\hbar^2 \Lambda^2}{2} \mathcal{M}_{2,+} \mathcal{M}_{2,-} \langle \Phi_{0,+} | \hat{\sigma}_x | \Phi_{0,-} \rangle,$$

$$\langle \hat{S}_y \rangle^2 = \left( 1 - \frac{\hbar^2 \Lambda^2}{4} \mathcal{M}_{0,-} \right) \langle \Phi_{0,-} | \hat{\sigma}_y | \Phi_{0,-} \rangle \tag{28b}$$

$$+ \frac{\hbar^2 \Lambda^2}{4} \left( \mathcal{M}_{2,+}^2 \langle \Phi_{0,+} | \hat{\sigma}_y | \Phi_{0,+} \rangle + \mathcal{M}_{2,-}^2 \langle \Phi_{0,-} | \hat{\sigma}_y | \Phi_{0,-} \rangle \right)$$

$$\langle \hat{S}_z \rangle^2 = \left( 1 - \frac{\hbar^2 \Lambda^2}{4} \mathcal{M}_{0,-} \right) \langle \Phi_{0,-} | \hat{\sigma}_z | \Phi_{0,-} \rangle + \frac{\hbar^2 \Lambda^2}{2} \mathcal{M}_{0,+} \langle \Phi_{0,+} | \hat{\sigma}_x | \Phi_{0,-} \rangle$$

$$+ \frac{\hbar^2 \Lambda^2}{4} \left( \mathcal{M}_{2,+}^2 \langle \Phi_{0,+} | \hat{\sigma}_z | \Phi_{0,+} \rangle + \mathcal{M}_{2,-}^2 \langle \Phi_{0,-} | \hat{\sigma}_z | \Phi_{0,-} \rangle \right) \tag{28c}$$

$$+ \frac{\hbar^2 \Lambda^2}{2} \mathcal{M}_{2,+} \mathcal{M}_{2,-} \langle \Phi_{0,+} | \hat{\sigma}_z | \Phi_{0,-} \rangle.$$

Here, the factors $\mathcal{M}_{n,\pm}$ stem from second order perturbation theory and read

$$\mathcal{M}_{0,-} = \frac{2 |\langle \Phi_{2,+} | \hat{\sigma}_z | \Phi_{0,-} \rangle|^2}{\left( E_-^{(0)}(0) - E_+^{(0)}(2) \right)^2} + \frac{2 |\langle \Phi_{2,-} | \hat{\sigma}_z | \Phi_{0,-} \rangle|^2}{\left( E_-^{(0)}(0) - E_-^{(0)}(2) \right)^2}, \tag{29a}$$

$$\mathcal{M}_{0,+} = \frac{2 \langle \Phi_{0,-} | \hat{\sigma}_z | \Phi_{2,-} \rangle \langle \Phi_{2,-} | \hat{\sigma}_z | \Phi_{0,-} \rangle}{\left( E_-^{(0)}(0) - E_+^{(0)}(2) \right) \left( E_-^{(0)}(0) - E_+^{(0)}(0) \right)}$$

$$+ \frac{2 \langle \Phi_{0,+} | \hat{\sigma}_z | \Phi_{2,+} \rangle \langle \Phi_{2,+} | \hat{\sigma}_z | \Phi_{0,-} \rangle}{\left( E_-^{(0)}(0) - E_-^{(0)}(2) \right) \left( E_-^{(0)}(0) - E_+^{(0)}(0) \right)}, \tag{29b}$$

$$\mathcal{M}_{2,+} = \frac{\sqrt{2} \langle \Phi_{2,+} | \hat{\sigma}_z | \Phi_{0,-} \rangle}{E_-^{(0)}(0) - E_+^{(0)}(2)}, \tag{29c}$$

$$\mathcal{M}_{2,-} = \frac{\sqrt{2} \langle \Phi_{2,-} | \hat{\sigma}_z | \Phi_{0,-} \rangle}{E_-^{(0)}(0) - E_-^{(0)}(2)}. \tag{29d}$$

The behavior of $|\langle \hat{\boldsymbol{S}} \rangle|$ within this approximation is depicted in Fig. 2(b) and (c).

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
