# Peer review of "Competition of light- and phonon-dressing in microwave-dressed Bose polarons"

_SciPost Physics_

## Round 1 · Referee Report · Arturo Camacho Guardian (Referee 1) · 2025-5-18

Strengths

  1. Solid theoretical framework which has extensively been used by the authors.
  2. Timely problem and interesting problem.
  3. Exhaustive study of their model.

Weaknesses

  1. Some aspects of the presentation can be improved.
  2. Not entirely clear what is the "theoretical/numerical " novelty of this study.

Report

The manuscript "Competition of light- and phonon-dressing in microwave-dressed Bose polarons" by G. Koutentakis and co-authors studies the quasiparticle properties of a spinor impurity coupled to a scalar BEC confined in a one-dimensional harmonic trap. The internal states of the impurity are coupled by a light-field, one of the states is interacting whereas the second one is non-interacting. The authors adapt their numerical approach ML-MCTDHX ( an ab-initio numerical approach) to study this system, this technique has been developed quite extensively in the group of some of the authors.

The topic of the manuscript is timely, the methodology robust, the results interesting and overall the manuscript is well-written. Before giving my final recommendations please find below my requested changes and questions.

Requested changes

  1. Some figures are quite crowded and difficult to interpret. For instance, in Fig. 1(a) and (b), it is hard to distinguish the differences between the plots, understand the role of N=2, and determine whether impurity-impurity interactions are present. The arrows in Fig. 1(b) are not particularly useful and add to the confusion.
  2. Perhaps I missed something, but one of the main conclusions seems to be that the residue increases with light-dressing. However, I couldn't find a figure that clearly demonstrates this. In fact, most of the results show a residue close to 1, so I struggle to understand how the impurity is dressed at all. (see next point)
  3. I also found the notation somewhat confusing, particularly in Section 3. My understanding is that the system can be described by a simple two-level model consisting of the non-interacting impurity state and the polaron state (in the limit of vanishing light-matter coupling). These two states then couple to the light field and hybridize. If this is correct, why is the residue not simply given by Eq. (7)? 4.My main concern relates to the novelty and significance of the results, which connects to the previous comment. It appears that the light-matter coupling can be described by a simple two-level system, and that, in the explored regime, no particularly intriguing phenomena emerge. From what I understand, the main observable effect comes from Fig. 1(d), where there is a “drop” in S from 0.5 to 0.492—this seems rather small. Could the authors comment on the physical relevance of such a small difference? Is it experimentally observable? Also, did the use of the ML-MCTDHX method require any non-trivial extensions for this study?
  4. On page 13, the authors refer to the 1/k^4 scaling related to Tan’s contact to support their claims. I don’t fully understand this argument. As far as I know (though I may be mistaken), this scaling is a high-energy feature usually captured only with non-perturbative methods. While I agree that ML-MCTDHX is an ab initio approach, it is unclear to me how it could include the relevant two-body physics (e.g., Feshbach resonance physics) necessary to reproduce the 1/k^4 scaling. Minor comments: On page 2, second paragraph: it reads quasiparticletheories — a space is missing. On page 2, the authors cite Refs. [69–82] as related to their previous work. Since this manuscript builds on those studies, could the authors clarify the relevance of each reference for the present work? They begin to do this on page 3, but the discussion could be made more explicit.

Recommendation

Publish (easily meets expectations and criteria for this Journal; among top 50%)

  • validity: high
  • significance: high
  • originality: good
  • clarity: high
  • formatting: excellent
  • grammar: excellent

Author:  Georgios M. Koutentakis  on 2025-07-21  [id 5657]

(in reply to Report 1 by Arturo Camacho Guardian on 2025-05-18)
Category:
answer to question
reply to objection

Dear Referee,

Please find attached our point by point answer to your suggestions and comments.

On behalf of the authors,
Georgios Koutentakis

Attachment:

Reply_report1_Scipost.pdf

---

## Round 1 · Referee Report · Anonymous (Referee 2) · 2025-5-28

Strengths

  1. Comprehensive theoretical analysis using different methods
  2. Interesting problem

Weaknesses

  1. Very long paper and sometimes confusing (or erroneous) notation
  2. Unclear what the main results and why they are interesting

Report

This manuscript considers the properties of one or a few bosonic impurities in a 1D weakly interacting Bose gas. The main novelty is that the impurity has two internal spin states, which are Rabi coupled: one internal state is interacting with the surrounding bosons and one is not. By carefully comparing with ab initio numerical calculations, the authors show that two simple analytical approaches based on a two-level approximation and an effective Hamiltonian both provide an accurate description of the system.

The topic of the manuscript is timely and interesting since there is presently a significant research activity exploring the physics of mobile impurities in a quantum degenerate environment. Before the paper is published, I however request the authors to consider the points below.

Requested changes

  1. Is it really correct to speak about polarons in 1D? The existence of quasiparticles in 1D is not obvious. For instance, in arXiv:2504.17558 it was shown that a mobile impurity in a Fermi gas has zero quasiparticle residue. I presume the same holds for an impurity in a Bose gas? I suggest the authors discuss this in the manuscript.

  2. In Fig. 1, the case of zero Rabi coupling is labeled as a solid black line. However, I think the line is dashed.

  3. I don't understand the formula for the residue Z given in the fourth line of page 8. Doesn't this formula give zero?

  4. The authors in general find very small deviations between the exact numerical calculations and the approximate theories. It would make the paper more interesting and suitable for SciPost, if the authors explore if this holds also for stronger impurity-boson interactions - especially on the attractive side where there is no phase separation. In particular, the authors state on page 14 that the effective Hamiltonian is accurate for g_BI<g_BB. Is this obvious for large and negative g_IB?

  5. On page 18, the authors state that detunings \Delta<-3\omega_B are suitable for studying the repulsive polaron. This probably depends on the value of g_IB. Should the condition involve the polaron energy instead?

Recommendation

Publish (meets expectations and criteria for this Journal)

  • validity: high
  • significance: ok
  • originality: good
  • clarity: ok
  • formatting: good
  • grammar: perfect

Author:  Georgios M. Koutentakis  on 2025-07-21  [id 5656]

(in reply to Report 2 on 2025-05-28)
Category:
answer to question
reply to objection

Dear Referee,

please find our response to your suggestions and comments in the attached pdf.

On behalf of the authors,
Georgios Koutentakis

Attachment:

Reply_report2_Scipost.pdf

---

## Editorial Decision

resubmitted